# Characterizing DNA methylation signatures of retinoblastoma using aqueous humor liquid biopsy

Hong-Tao Li [1,8], Liya Xu [2,8], Daniel J. Weisenberger [3,4], Meng Li[5], Wanding Zhou[6], Chen-Ching Peng[2], Kevin Stachelek[2], David Cobrinik[2,3,4,7], Gangning Liang [1,4] ✉ & Jesse L. Berry [2,4,7] ✉

Retinoblastoma (RB) is a cancer that forms in the developing retina of babies and toddlers. The goal of therapy is to cure the tumor, save the eye and maximize vision. However, it is difficult to predict which eyes are likely to respond to therapy. Predictive molecular biomarkers are needed to guide prognosis and optimize treatment decisions. Direct tumor biopsy is not an option for this cancer; however, the aqueous humor (AH) is an alternate source of tumor-derived cell-free DNA (cfDNA). Here we show that DNA methylation profiling of the AH is a valid method to identify the methylation status of RB tumors. We identify 294 genes directly regulated by methylation that are implicated in p53 tumor suppressor (RB1, p53, p21, and p16) and oncogenic (E2F) pathways. Finally, we use AH to characterize molecular subtypes that can potentially be used to predict the likelihood of treatment success for retinoblastoma patients.

Retinoblastoma (RB) is a childhood cancer that forms in the developing retina of babies and toddlers. Malignant tumors can form in one or both eyes. Without treatment RB is life threatening; even with treatment there is a high chance of losing the eye if cancer does not respond to therapy[1,2]. Although the majority of RB cases are initiated by biallelic inactivating mutations of the *RB1* tumor suppressor gene[3], ~13% of non-hereditary RB have *RB1* promotor methylation and silencing[4–6]. In addition, epigenetic deregulation of tumor-promoting pathways has been shown to be important for RB tumorigenesis and disease progression beyond *RB1* inactivation[7–9]. Due to the fundamental role epigenetics plays in this malignancy, a mechanism to assay epigenetic tumor profiles in vivo from eyes undergoing salvage therapy would be highly relevant.

While tumor biopsy is the diagnostic norm for most malignancies, direct tissue biopsy is contraindicated for RB due to the risk of provoking extraocular tumor spread. Until recently, this contraindication meant that no molecular tumor information was available unless the eye was enucleated (surgically removed). However, in 2017 we demonstrated that the aqueous humor (AH), the clear fluid in front of the eye, is an enriched source of tumor-derived cell-free DNA (cfDNA) for RB[10,11], that facilitates analysis of tumor-derived cfDNA in the absence of tumor tissue. Molecular genomic profiling of AH cfDNA opens the door to apply decades of knowledge about RB genomics in an impactful in vivo clinical application[10,12–14]. The addition of epigenetic assays enables a better understanding of the role of methylation in orchestrating gene expression in disease initiation and progression.

[1]Department of Urology, University of Southern California, Norris Comprehensive Cancer Center, Los Angeles, CA 90033, USA. [2]Children's Hospital Los Angeles Vision Center & USC Roski Eye Institute, Keck School of Medicine, University of Southern California, Los Angeles, CA 90027, USA. [3]Department of Biochemistry and Molecular Medicine, University of Southern California, Norris Comprehensive Cancer Center, Los Angeles, CA 90033, USA. [4]Norris Comprehensive Cancer Center, Keck School of Medicine, University of Southern California, Los Angeles, CA 90033, USA. [5]Norris Medical Library, University of Southern California, Los Angeles, CA 90033, USA. [6]University of Pennsylvania, Children's Hospital of Philadelphia, Philadelphia, PA 19104, USA. [7]The Saban Research Institute, Children's Hospital Los Angeles, Los Angeles, CA 90089, USA. [8]These authors contributed equally: Hong-Tao Li, Liya Xu. ✉e-mail: gliang@usc.edu; jesse.berry@med.usc.edu

This includes the identification of tumors initiated by DNA hypermethylation of the *RB1* or other gene promoters that may help stratify patients for epigenetic treatment regimens. Thus, epigenetic analysis of AH cfDNA is a highly desired aspect of an integrated, multi-modal liquid biopsy platform.

In this study, we performed genome-scale DNA methylation profiling of paired AH cfDNA and primary RB tumors and integrated the results with existing RB tumor DNA methylation profiles. The methylation profiles of AH cfDNA and primary tumors show high concordance, demonstrating that the AH profiling is a reliable mechanism to evaluate the methylation status of the tumor. We further performed enriched pathway analysis to identify aberrantly methylated genes directly involved in RB tumorigenesis. Finally, we demonstrated the ability to identify *RB1* promoter DNA hypermethylation, a known cause of sporadic, non-heritable RB, as well as DNA methylation profiles that may predict an aggressive tumor subtype less likely to respond to medical therapy. Our findings support accessible approaches of molecular-based RB diagnosis, and potential future clinical implications of epigenetic dysregulation in RB using this liquid biopsy.

## Results

### Validation of DNA methylation profiles in RB specimens

Genome-scale DNA methylation profiling of AH cfDNA and RB tumors was investigated to characterize RB epigenetic changes in vivo. DNA methylation profiles of four paired primary RB tumors and AH cfDNAs (CHLA 1–4) were measured using the Illumina MethylationEPIC (EPIC) DNA methylation BeadArray system. An additional 11 AH cfDNA (CHLA 5–15) samples collected at diagnosis or at the time of enucleation (i.e., surgical removal of the eye) were similarly analyzed (Table 1). DNA methylation datasets were filtered as per standard to remove data from probes that are: (1) linked to known polymorphisms, (2) located on the X- and Y-chromosomes, and (3) related to aging (Fig. S1). Publicly available DNA methylation data (Illumina Infinium HumanMethylation450, HM450) for primary RB tumors (RB_SR) ($n = 57$) and tumor-adjacent retinas ($n = 12$) as controls (GSE57362)[9] were integrated for validation. By overlapping our EPIC array data with the published HM450 data, a total of 363,579 probes remained for downstream analysis. After filtering for RB tumor purity, 34 primary RB (RB_SR_1–30 and RB_CHLA_1–4) and 15 AH cfDNA (AH_CHLA_1–15) samples were retained for further analysis.

We sought to identify RB tumor-specific DNA methylation changes. Welch's *t*-test was applied on the filtered 363,579 probe set to identify differentially methylated probes across the four primary RB samples from CHLA and the 30 tumors and 12 retinas from a publicly available dataset. With average β value difference >0.3 and $p < 0.05$, 15,483 probes were identified that are significantly differentially methylated between retina and RB samples. DNA methylation changes were identified in 31 of the 34 RB tumors with three exceptions (RB SR_18, 21, and 29) that displayed DNA methylation profiles similar to the normal retina (Fig. 1A). Approximately 19% of the probes showed strong DNA hypermethylation in RB samples, while 81% displayed DNA hypomethylation, consistent with the previous reporting[8].

Similarly, multidimensional scaling (MDS) of the DNA methylation data revealed that RB tumors mainly clustered separately from normal retina, aside from the three aforementioned tumors (RB_SR_18, 21, and 29) that may represent uninvolved retina (Fig. 1B). The MDS analysis also revealed greater DNA methylation heterogeneity in RB tumors versus the normal retina (Fig. 1A, B).

DNA hyper- or hypomethylation occurs in promoters, gene bodies, enhancer elements, and other as inter-genetic region. However, in the RB-specific group of 15,483 probes, DNA hypermethylated loci were mostly enriched within gene bodies and DNA hypomethylation was most prevalent in gene promoter regions (Fig. 1C). These findings

are suggestive of gene expression alterations, as promoter DNA hypomethylation and gene body DNA hypermethylation are correlated with gene activation[15,16]. In addition, DNA methylation in intergenic regions may correlate with chromatin instability and regulation of functional elements, such as enhancers[17,18]. The distribution of RB-specific DNA methylation alterations across various genic regions may provide clues regarding potential gene activity.

### DNA methylation profiles in cfDNA of aqueous humor (AH) are reliably assayed and highly concordant to primary RB tumors

The AH cfDNA, like that of other body fluids, is highly fragmented[19]. The Illumina Infinium EPIC DNA Methylation BeadChip is a widely used genome-scale DNA methylation assay[20], however, applying this technology for DNA methylation profiling of highly degraded, low input DNA samples, such as FFPE-DNA or cfDNA with less than the recommended input DNA amounts (250 ng), presented a challenge.

We evaluated the lower limits of fragmented DNAs on the EPIC DNA methylation array using short DNA fragments. Genomic DNA extracted from the human HCT116 colon cancer cell line was first sonicated to 200–300 bp to match AH cfDNA fragments and then 1, 5, 10, and 20 ng of the fragmentized DNA were subjected to the Illumina Restoration Kit after bisulfite conversion which is recommended for repairing FFPE-DNA samples prior to hybridization to Illumina EPIC DNA methylation arrays. 200 ng DNA was used as a control for bulk DNA amounts commonly evaluated on the EPIC DNA methylation array platform.

DNA methylation β values for 1, 5, 10, and 20 ng of the repaired DNA were plotted versus the 200 ng DNA sample (Fig. 2A). The 1 ng input DNA sample showed some DNA methylation β value skewing compared to the bulk sample, but still showed a high correlation ($r^2 = 0.899$) to the bulk 200 ng DNA sample (Fig. 2B). The scatterplots and associated correlation coefficients show a strong and reliable association of the 5, 10, 20 ng input DNA samples vs. the bulk 200 ng control sample ($r^2 = 0.97$–$0.98$) (Fig. 2B). These findings suggested that Illumina EPIC DNA Methylation assay is applicable for measuring DNA methylation of cfDNA samples with >1 ng input DNA and as such can be applied to the lower amount of cfDNA in the AH.

DNA methylation data on AH cfDNA samples with 1–10 ng input was successfully generated. DNA methylation profiles of four pairs of RB tumors and AHs (CHLA 1–4) demonstrated highly concordant DNA methylation profiles for each tumor-AH pair and distinct separation between different tumor-AH pairs (Fig. 2C). Unsupervised clustering of the most variably methylated probes across all four tumor-AH pairs also highlighted differential DNA methylation among these four patients and highly concordant DNA methylation profiles between each RB tumor and its corresponding paired AH (Fig. 2D); this demonstrates that the AH could be used in the absence of tumor (e.g., from eyes that have not been surgically removed) to accurately assay the methylation signature of the tumor in vivo. Copy number analysis of the EPIC DNA methylation data set[18,21] also showed high concordance between each primary RB tumor and its paired AH cfDNA specimen (Supplementary Fig. 3A).

An additional set of 11 AH samples from CHLA were included for further comparison, such that a total of $n = 15$ AH cfDNA specimens were analyzed. All 15 AH_CHLA samples showed the RB-specific DNA methylation pattern by unsupervised clustering (Fig. 3A) and MDS (Fig. 3B) analyses. The RB-specific DNA methylation profiles were not detected in cfDNA isolated from blood plasma in two RB patients but rather clustered with white blood cell DNA isolated from RB patients (Fig. S4) as expected since the disease was confined to the eye without high tumor fraction in the blood[12,22].

Analyses of gene-level DNA methylation in Fig. 3A revealed *RB1* promoter DNA hypermethylation in five samples (RB_CHLA_3, AH_CHLA_3, RB_SR_16, RB_SR_24, and RB_SR_30) consistent with

**Table 1 | Clinical demographics, diagnostic clinical features, and AH sampling time for each study participant**

| Sample ID | Sample type | EPIC array barcode | Study_ID | Case ID | Sex | Age at Dx (mos) | IIRC group | TNM stage | Laterality | Germline mutation | Initial Tx | AH sampling time | Tumor availability | Outcome |
|---|---|---|---|---|---|---|---|---|---|---|---|---|---|---|
| RB_CHLA_1 | Primary tumor | 205549600103_R05C01 | CHLA_RR_030_OD_Tumor | 49 | M | 35 | D | cT1b | U | *RB1* negative | ENUC | PE | Yes | Primary enucleated |
| RB_CHLA_2 | Primary tumor | 205549600103_R06C01 | CHLA_RR_037_OD_Tumor | 54 | M | 25 | E | cT3c | U | *RB1* negative | ENUC | PE | Yes | Primary enucleated |
| RB_CHLA_3 | Primary tumor | 205549600103_R07C01 | CHLA_RR_045_OD_Tumor | 57 | F | 24 | E | cT3c | U | *RB1* negative | ENUC | PE | Yes | Primary enucleated |
| RB_CHLA_4 | Primary tumor | 205549600103_R08C01 | CHLA_RR_050_OS_Tumor | 65 | M | 24 | D | cT2b | U | APC: c.3920 T > A | ENUC | PE | Yes | Primary enucleated |
| AH_CHLA_1 | Aqueous humor | 205549600103_R01C01 | CHLA_RR_030_OD | 49 | M | 35 | D | cT1b | U | *RB1* negative | ENUC | PE | Yes | Primary enucleated |
| AH_CHLA_2 | Aqueous humor | 205549600103_R02C01 | CHLA_RR_037_OD | 54 | M | 25 | E | cT3c | U | *RB1* negative | ENUC | PE | Yes | Primary enucleated |
| AH_CHLA_3 | Aqueous humor | 205549600103_R03C01 | CHLA_RR_045_OD | 57 | F | 24 | E | cT3c | U | *RB1* negative | ENUC | PE | Yes | Primary enucleated |
| AH_CHLA_4 | Aqueous humor | 205549600103_R04C01 | CHLA_RR_050_OS | 65 | M | 24 | D | cT2b | U | APC: c.3920 T > A | ENUC | PE | Yes | Primary enucleated |
| AH_CHLA_5 | Aqueous humor | 205624900017_R02C01 | CHLA_RR_022_OS | 44_OS | F | 4 | D | cT2b | B | *RB1*:c.1666C > T | CEV | DX | No | Salvaged |
| AH_CHLA_6 | Aqueous humor | 205624900017_R04C01 | CHLA_RR_048_OD | 64_OD | F | 15 | D | cT2b | B | *RB1*: deletion Ex24-27 | CEV | DX | No | Salvaged |
| AH_CHLA_7 | Aqueous humor | 205624900017_R05C01 | CHLA_RR_048_OS | 64_OS | F | 15 | E | cT2b | B | *RB1*: deletion Ex24-27 | CEV | ES | No | Secondary enucleated |
| AH_CHLA_8 | Aqueous humor | 205648300012_R03C01 | CHLA_RR_024_OD | 45 | F | 8 | D | cT2b | U | *RB1* negative | CEV | DX | No | Salvaged |
| AH_CHLA_9 | Aqueous humor | 205648300012_R04C01 | CHLA_RR_025_OS | 46 | M | 5 | C | cT2a | U | *RB1* negative | CEV | DX | No | Salvaged |
| AH_CHLA_10 | Aqueous humor | 205624900017_R03C01 | CHLA_RR_028_OD | 47 | F | 15 | D | cT2b | U | *RB1* negative | CEV | DX | No | Salvaged |
| AH_CHLA_11 | Aqueous humor | 205648300012_R05C01 | CHLA_RR_041_OS_DX | 55 | M | 24 | D | cT2b | U | *RB1* negative | IAC | DX | No | Secondary enucleated |
| AH_CHLA_12 | Aqueous humor | 205648300012_R06C01 | CHLA_RR_041_OS_ES | 55 | M | 24 | D | cT2b | U | *RB1* negative | IAC | ES | No | Secondary enucleated |
| AH_CHLA_13 | Aqueous humor | 205648300012_R01C01 | CHLA_RR_023_OS_DX | 33 | M | 22 | D | cT2b | U | *RB1* negative | IAC | DX | No | Secondary enucleated |
| AH_CHLA_14 | Aqueous humor | 205648300012_R02C01 | CHLA_RR_023_OS_ES | 33 | M | 22 | D | cT2b | U | *RB1* negative | IAC | ES | No | Secondary enucleated |
| AH_CHLA_15 | Aqueous humor | 205624900017_R07C01 | 2016_021_OD | 10 | F | 2 | E | cT3c | B | *RB1*: c.2425delC | CEV | ES | No | Secondary Enucleated |
| Blood_CHLA_1 | Blood plasma | 205624900017_R06C01 | RR_022_DX_Blood_cfDNA | 44 | F | 4 | D | cT2b | B | *RB1*:c.1666C > T | CEV | DX | No | Salvaged |
| Blood_CHLA_2 | Blood plasma | 205624900017_R08C01 | RR_48_ES_Blood_cfDNA | 64 | F | 15 | E | cT2b | B | *RB1*: deletion Ex24-27 | CEV | ES | No | Secondary enucleated |

*OD right eye, OS left eye, M male, F female, IIRC International Intraocular Retinoblastoma Classification, U unilateral, B bilateral, PE primary enucleation, DX diagnostic, SE secondary enucleation, RB1 retinoblastoma tumor suppressor gene.*

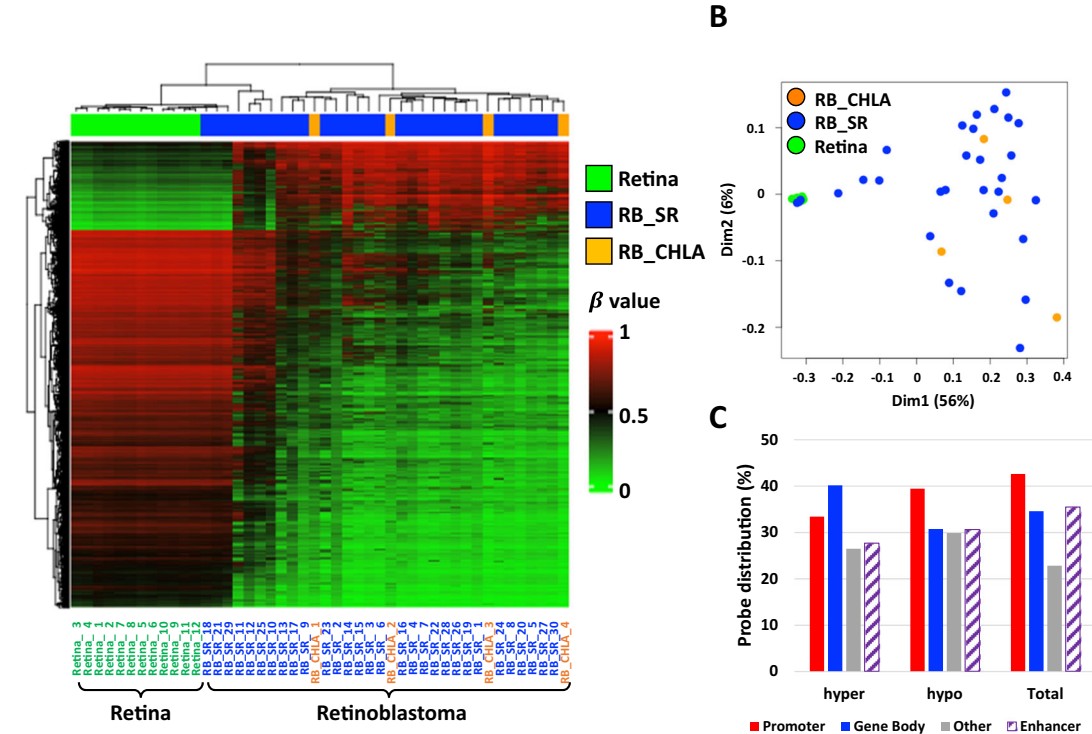

**Fig. 1 | DNA methylation signatures in retinoblastoma (RB). A** Unsupervised hierarchical clustering of the differentially methylated probes between RB and retina samples. The top annotation indicates sample type. Green and blue bars indicate the retina (green, $n = 12$) and RB samples (RB_SR, $n = 30$), respectively, from dataset GSE57362. Orange bars highlight the four primary RB tumor samples collected for this study (RB_CHLA, $n = 4$). **B** MDS plot showing that RB samples ($n = 34$) can be distinguished from retina ($n = 12$) using the panel of differentially methylated probes in **A**. **C** Genomic location of the hyper- and hypo- CpGs on promoter (TSS, 5'UTR, and 1st exon), gene body, other and enhancer. Total means the probe distribution on EPIC array. Source data are provided as a Source Data file.

*RB1* inactivation via epigenetic silencing; a known mechanism of non-heritable *RB1* inactivation. Previously this could only be determined with access to tumor tissue, however identical *RB1* promoter DNA hypermethylation was detected in cfDNA of AH and from tumor DNA from the same enucleated eye (RB_CHLA_3 and AH_CHLA_3) further demonstrating that *RB1* promoter DNA hypermethylation can be reliably detected via ocular AH liquid biopsy in the absence of tumor tissue (Fig. 3C).

Overexpression and/or amplification of *MYCN* and *SYK* have been demonstrated to highly correlated to RB tumorigenesis and considered as potential therapeutic targets[23–25]. In our cohort, the majority of RB tumors demonstrated *MYCN* and *SYK* promoter DNA hypomethylation (associated with gene activation); this profile was similarly identified in all AH samples as compared to apparently normal retinal tissues (Fig. 3C), suggesting that these targets can be detected via AH methylation profiling.

**Characterization of genes with RB-associated DNA methylation profiles and their involvement in RB tumorigenesis**
While most cancer-specific DNA methylation alterations do not result in altered gene expression[16,26,27], promoter DNA methylation is negatively correlated with gene expression and gene body DNA methylation is positively associated with gene expression[15,28]. To characterize the extent to which promoter and gene body DNA methylation affect gene expression, we used publicly available RNA sequencing (RNA-seq) data (GSE125903 and GSE111168)[29,30] due to the limited availability of our primary RB samples.

First, we identified DNA methylation probes exhibiting RB-specific DNA methylation changes (delta β value > 0.3) in promoter or gene body regions by comparing normal retina and primary RB samples (Fig. 1). In total, we identified promoter DNA hypermethylation (978 probes), promoter DNA hypomethylation (4949 probes), gene body DNA hypermethylation (1178 probes) and gene body DNA hypomethylation (3856 probes) (Fig. 4A). In addition, we uncovered upregulation of 889 and downregulation of 382 genes in RB. After integrating the RB-specific promoter and gene body DNA methylation (Fig. 4A) with differential gene expression data, we identified 294 genes that show potential gene regulation by aberrant DNA methylation directly in RB (Fig. 4B). These genes include those upregulated and correlated with promoter DNA hypomethylation ($n = 172$) or gene body DNA hypermethylation ($n = 37$), as well as those downregulated and correlated with promoter DNA hypermethylation ($n = 67$) or gene body DNA hypomethylation ($n = 18$) (Fig. 4B and Supplementary Data 1). Although Illumina-based DNA methylation data are reliable and have been validated using pyrosequencing and targeted bisulfite sequencing[27,31], we confirmed the EPIC DNA methylation array data by performing targeted bisulfite sequencing of four gene regions (*TFF1* and *HOXC4* promoters and *MNX1* and *CELSR3* gene bodies) on 10 additional primary RB tumors and one apparently healthy retina (Supplementary Data 3 and Supplementary Fig. 5). The EPIC array and targeted bisulfite sequencing DNA methylation data are highly consistent at these four loci.

The potential roles of epigenetic-directed gene expression during RB tumorigenesis remain unclear. Core analysis in Ingenuity Pathway Analysis (IPA, Qiagen) was performed on the set of 294 differentially expressed genes to understand their potential functional profiles for RB tumorigenesis (Fig. 4B, C). The Graphical Summary algorithm predicted down-regulation of tumor suppressor pathways involving p53, RB1, CDKN2A/p16, and CDKN1A/p21, and activation of oncogenic pathways involving E2F1, E2F2, E2F3, and MYC (Fig. 4C). Furthermore, we used TRANSFAC analysis to identify transcription factors (TFs) involved in the regulation of these pathways or genes[32]. The top overrepresented transcription factors binding sites included oncogenic regulators involved in ER (Estrogen receptor), Ras (RREB-1, Ras

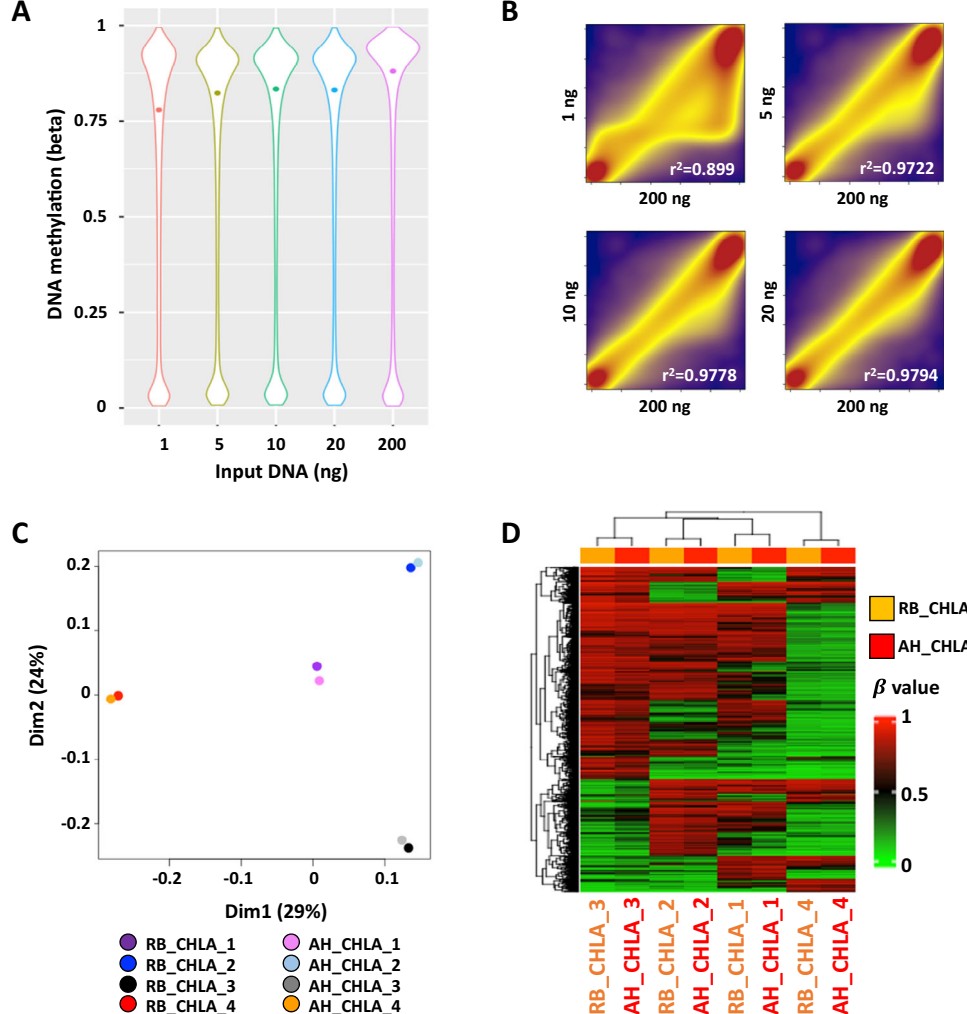

**Fig. 2 | DNA methylation profiles in AH cfDNAs are highly concordant with paired primary RB tumors. A** Violin plot showing the global DNA methylation level of sonicated DNA from the HCT116 cell line. DNA input amounts for EPIC methylation array assays are shown at the bottom. Non-sonicated bulk DNA (200 ng) was used as control. The median methylation values are shown in dots for each sample. **B** Smooth Kernel scatter plots showing the DNA methylation patterns of different amounts of sonicated DNA input DNA (y-axis) compared to the bulk DNA sample (x-axis). The $r^2$ values are displayed for each plot. **C** MDS plot indicating the concordance of individual CHLA RB primary tumor sample (RB_CHLA, $n = 4$) and the paired AH cfDNA sample (AH_CHLA, $n = 4$) demonstrating AH is a valid substrate to assay tumor methylation signature. Top 10,000 most variably methylated probes were used to generate the plot. **D** Unsupervised hierarchical clustering analysis and heatmap representation of the methylation of the probes used in panel **C**.

responsive element binding protein 1), E2F, MYC (MAZ, MYC associated zinc finger protein), NF-kB, and EGR1 (Early growth response protein 1) signaling pathways (Supplementary Fig. 6).

These pathways are known to be involved in RB tumorigenesis and contribute to upregulation of aryl hydrocarbon receptor (AhR) signaling, estrogen-mediated S-phase entry for cancer cell proliferation, and others[33–35]. These findings suggest that in addition to genetic alterations such as pathogenic *RB1* variants, DNA methylation-regulated genes related to cancer aggressiveness can be involved in the downregulation of tumor suppressor pathways as well as upregulation of oncogenic pathways that contribute to RB tumorigenesis. Further detailed analysis of Estrogen-mediated S-phase pathway elements revealed several key downstream signaling genes that are upregulated in association with promoter DNA hypomethylation in RB tumors, including *CCNA1* and *CCNA2* for Cyclin A, *CCNE1* and *CCNE2* for Cyclin E, as well as *E2F1, E2F2,* and *CDC2* (Fig. 4D). Interestingly, our data also showed that these genes, especially in downstream of RB1 such as Cyclin A, Cyclin E, and CDC2, can be upregulated by promoter DNA hypomethylation independent of germline *RB1* mutation status (Fig. 4E).

## AH cfDNA methylation profiles are associated with RB tumor aggressiveness

To explore the potential of epigenomic prognostic biomarkers to predict eye salvage via AH, we analyzed EPIC DNA methylation data from 12 AHs from eyes with different clinical outcomes, including four eyes salvaged with therapy (SV) with at least 1 ng AH cfDNA (AH_CHLA_6 removed), four primarily enucleated eyes (PE) without initial medical intervention and four secondary enucleations (SE) in which the eye failed attempted treatment. Three AH samples with low data quality were excluded. Specifically, AH cfDNA methylation profiles between salvage (AH_CHLA_5, 8, 9, and 10), primary enucleation (AH_CHLA_1, 2, 3, and 4) and secondary enucleation (AH_CHLA_11, 12, 13, and 14) cases were analyzed using heatmap representation after unsupervised clustering. In total, 1092 probes were identified that are significantly differentially methylated ($\triangle\beta > 0.4$, $p < 0.01$) between the salvage ($n = 4$) and enucleation groups ($n = 4$ PE and $n = 4$ SE) (Fig. 5A). As expected, based on our previous work[12] and that of Liu et al.[36], salvaged eyes had fewer copy number alterations than enucleated eyes, especially for gain of 1q, 6p and loss of 16q in current dataset (Supplementary Fig. 3B).

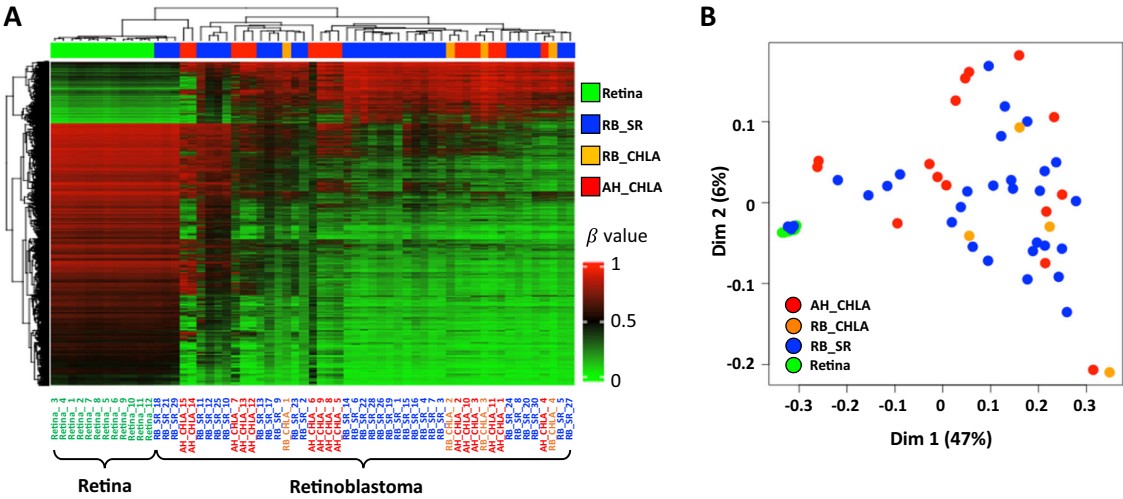

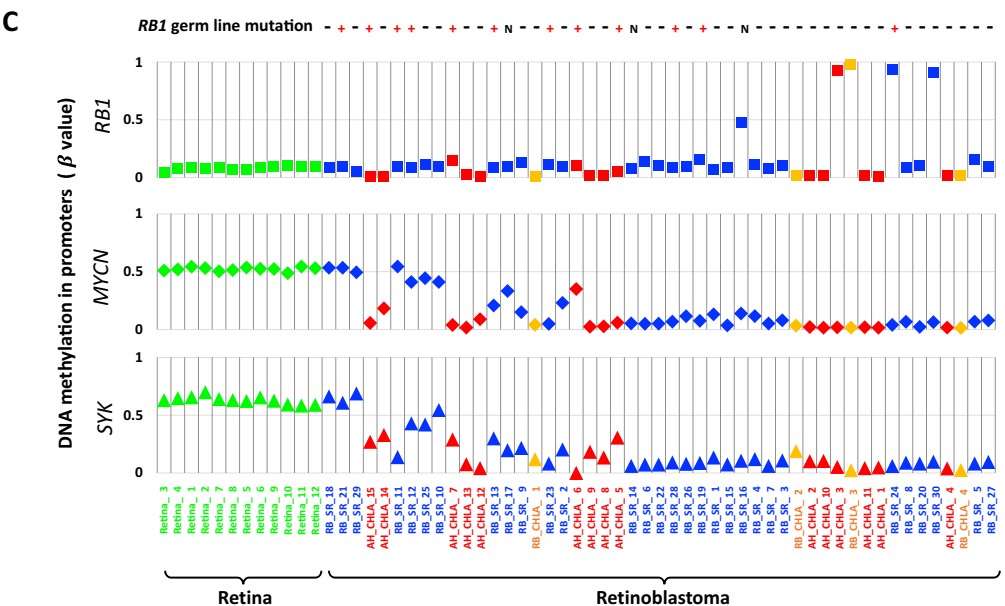

**Fig. 3 | RB-specific DNA methylation profiles are highly correlated in AH and primary tumors. A** Unsupervised hierarchical clustering of differential DNA methylation between retina (n = 12) and RB samples including AH cfDNA (n = 15) and RB tumor samples (n = 34). The probes identified in Fig. 1A were used. **B** MDS plot showing AH cfDNA samples (n = 15) are clustered with primary RB tumor samples (n = 34) and separated from retina (n = 12). **C** Promoter DNA methylation *RB1*, *MYCN*, and *SYK* genes in normal retina and RB samples. The *RB1* germ line mutation status for each sample is shown on the top. "+" indicates *RB1* mutation, "-" indicates no *RB1* mutation, and "N" indicates that the genotype was not determined. This demonstrates that the AH can be used to identify cases of RB driven by *RB1* hypermethylation (CHLA_3, AH and tumor); additionally, *SYC* and *MYCN* promotors are hypomethylated as compared to normal retina suggesting gene activation in RB tumors versus normal retina. Source data are provided as a Source Data file.

To determine if these same probes could be used to distinguish primary tumors, DNA methylation data for this set of 1092 differentially methylated probes was further applied to the above-described 30 primary RB tumors (RB_SR, GSE57362). Interestingly, in the larger data set there remained differential methylation between the salvaged group and the enucleated tumors (Fig. 5B). For further validation, this comparison was repeated with a second DNA methylation data set of 67 primary RB tumor samples (RB_NC, GSE58783)[36] using unsupervised clustering (Fig. 5C). The distinguishable pattern (Fig. 5A) between salvaged samples and enucleation samples in DNA from primary tumors or AH liquid biopsy (Fig. 5B, C) suggests that DNA methylation analyses of AH cfDNA samples from RB eyes may be used to predict eye salvage.

To further investigate if the AH cfDNA methylation signature discriminates distinct tumor subgroups, we performed unsupervised

clustering of the merged datasets: 4 AH cfDNAs collected at diagnosis from eyes that were salvaged, 8 AH cfDNAs from eyes that were eventually enucleated, and 97 (30 + 67) enucleated samples from primary RB tumors (Fig. 5D). While the heatmap representation after unsupervised clustering (a) and MDS analyses (b) demonstrate the variability of methylation signatures in RB tumors, there were two unique patterns on the edges of the spectrum. We identified a subset of tumors that had a similar methylation signature to CHLA salvaged tumors (Cluster A) and an opposite signature more typical of the tumors enucleated at CHLA, either primarily or secondarily (Cluster B). The distribution of salvaged tumors on one arm and subsequently enucleated tumors on the other arm was significant (p < 0.01). We further identified 320 significantly differentially methylated probes ($\triangle \beta > 0.4$, p < 0.01) spanning 185 unique genes between the Cluster A and Cluster B (Supplementary Data 2) that were well separated using MDS analysis in (c) of Fig. 5D.

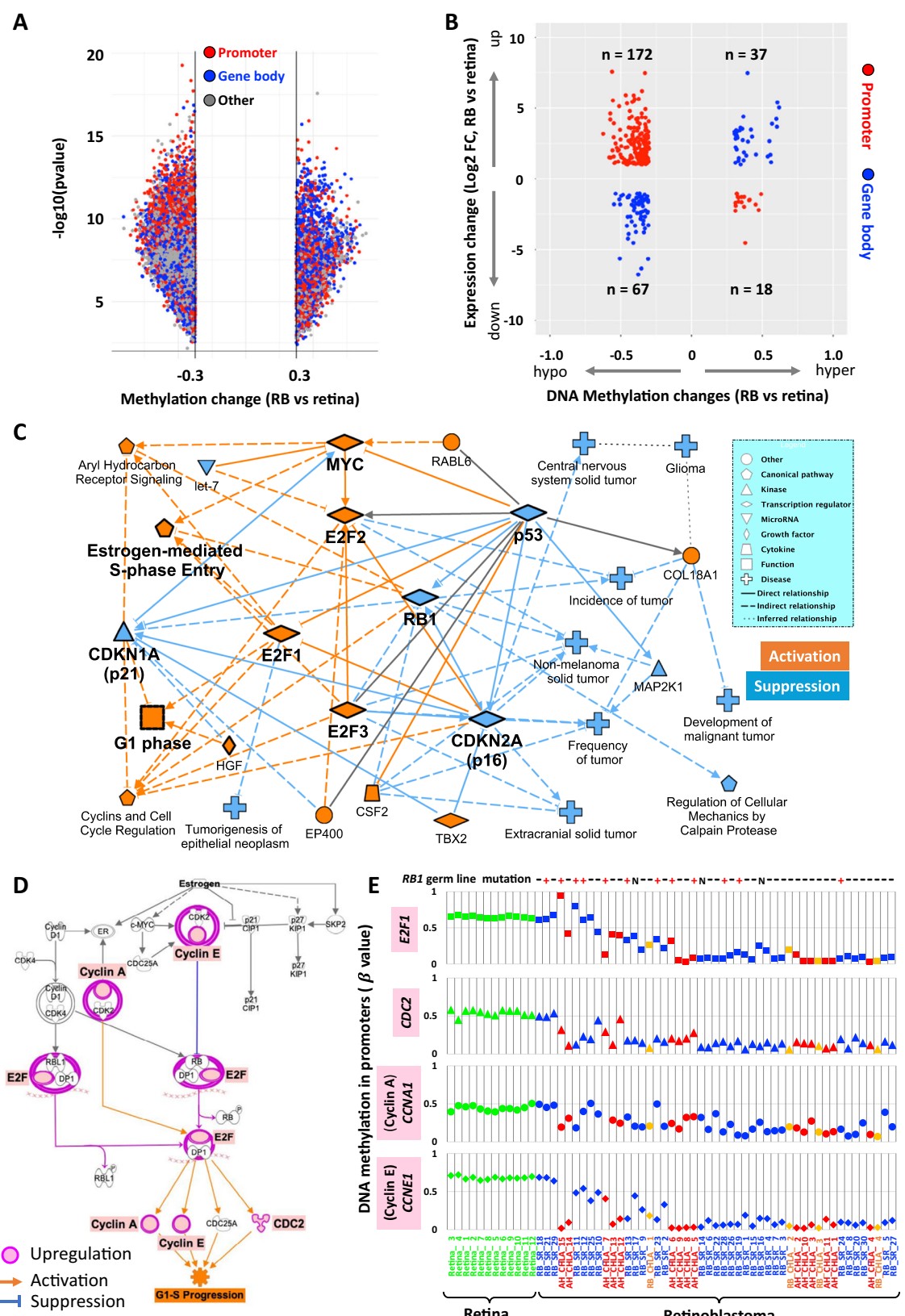

To better understand the characteristics of Cluster A and Cluster B tumors, we further investigated the data (GSE58783) from Liu et al.[36], who identified two RB tumor subtypes (Subtype 1 and Subtype 2) based on DNA methylation, copy number variation and gene expression profiles from 67 enucleated RB samples with DNA methylation data. Subtype 1 RB tumors maintained a differentiated state, while Subtype 2

RB tumors displayed more aggressive disease that is associated with dedifferentiation, stemness features, and expression of neuronal markers[36]. We applied the Subtype assignments with our Cluster A and B subgrouping, which were distinguished by 320 differentially methylated probes in AH cfDNA (Fig. 6A, B) and compared our assignments with theirs. Cluster A tumors ($n = 19$) fully overlapped with

**Fig. 4 | Pathway analyses of DNA methylation genes in RB. A** Significantly differential DNA methylation in RB versus retina. Each dot represents one probe. $-\log_{10}(p$ value$)$ for each probe were plotted on the y-axis while the β value difference between RB tumors and normal retina were plotted on the x-axis. The P value was calculated using two-sided Welch's t-tests. The β value change cutoffs of +/−0.3 were shown. Probe locations were shown in red for promoter, blue for gene body and gray for other location. **B** Genes regulated by DNA methylation in RB versus normal retina. DNA methylation changes were plotted on the x-axis (Δβ > 0.3 or <

−0.3). Gene expression changes were plotted on the y-axis (Log2FC > 1 or <−1). Only the probes located in promoter (red) and gene body (blue) were plotted. **C** Graphical summary of the IPA pathway analysis of the genes identified in **B** shows enrichment of several tumor-associated pathways. Source data are provided as a Source Data file. **D** The estrogen-mediated S-phase Entry pathway is activated by the upregulation of many pathway components (shown in red) identified in **B**. **E** Promoter DNA hypomethylation of the genes highlighted in **D**.

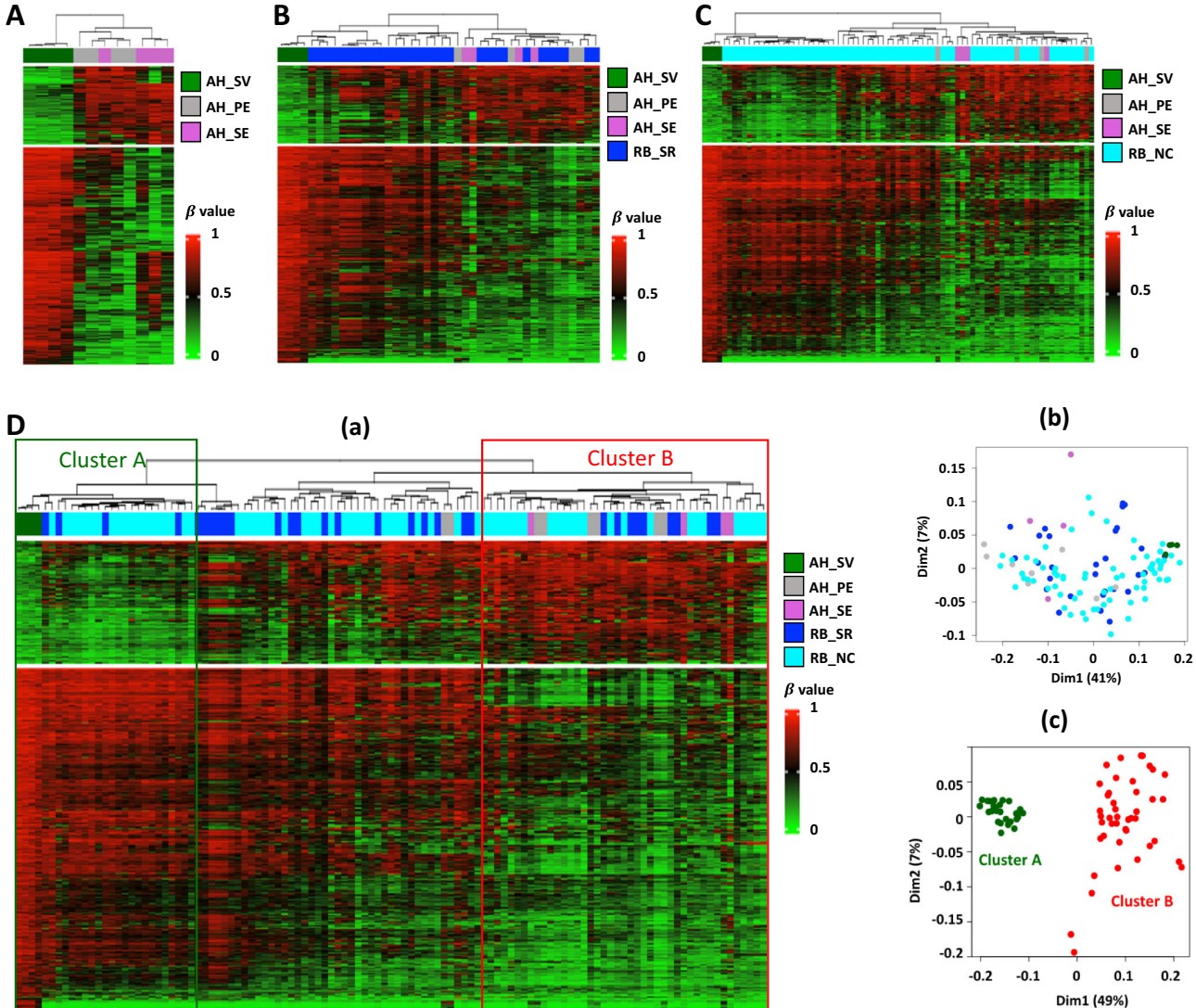

**Fig. 5 | Identification of specific DNA methylation clusters from AH cfDNAs.**
**A** Unsupervised hierarchical clustering of the 1092 differentially methylated probes between salvaged (AH_SV, dark green on top, $n=4$) and enucleated (AH_PE, primary enucleated ($n=4$) and AH_SE, secondary enucleated ($n=4$), gray and magenta on top) samples (top panel). The bottom panel showed the salvaged and enucleated samples can be well separated by the selected probes. **B** Unsupervised hierarchical clustering heatmap of CHLA AH samples ($n=12$) and enucleated RB tumors ($n=30$)
from GSE57362 cohort (RB_SR, blue) using the 1092 probes in **A**. **C** Unsupervised hierarchical clustering heatmap of the CHLA AH samples ($n=12$) and enucleated RB tumors ($n=67$) from GSE58783 cohort (RB_NC, aqua) using the 1092 probes in **A**. **D** Identification of Cluster A (less aggressive) and Cluster B (more aggressive) by the 1092 probes in **A**. (a) Unsupervised clustering of the 12 CHLA AH samples and 97 enucleated RB tumors (from **B** and **C**). (b) MDS plot of all the 109 (12 + 30 + 67) samples. (c) MDS plot of the Cluster A (green) and Cluster B (red) samples.

differentiated Subtype 1 ($n=27$) cases while Cluster B tumors ($n=24$) also fully overlapped with de-differentiated Subtype 2 ($n=37$) cases as described by unsupervised clustering and Venn diagram (Fig. 6A, B), suggesting our classifier for treatment outcome prediction is consistent with their classifier for disease aggressiveness. Taken together, these data suggested that a more aggressive RB tumor subtype could be predicted by cfDNA methylation profiles of AH liquid biopsies.

We performed a literature search to identify known oncogenesis genes that display the greatest extent of differential DNA methylation from the identified 185 genes (Supplementary Data 2) based on 320 differentially methylated probes between Clusters A and B. The examples of those genes that had either DNA hypomethylation in Cluster B tumors as compared to Cluster A tumors (Fig. 6C), or DNA hypermethylation in Cluster B tumors as compared to Cluster A

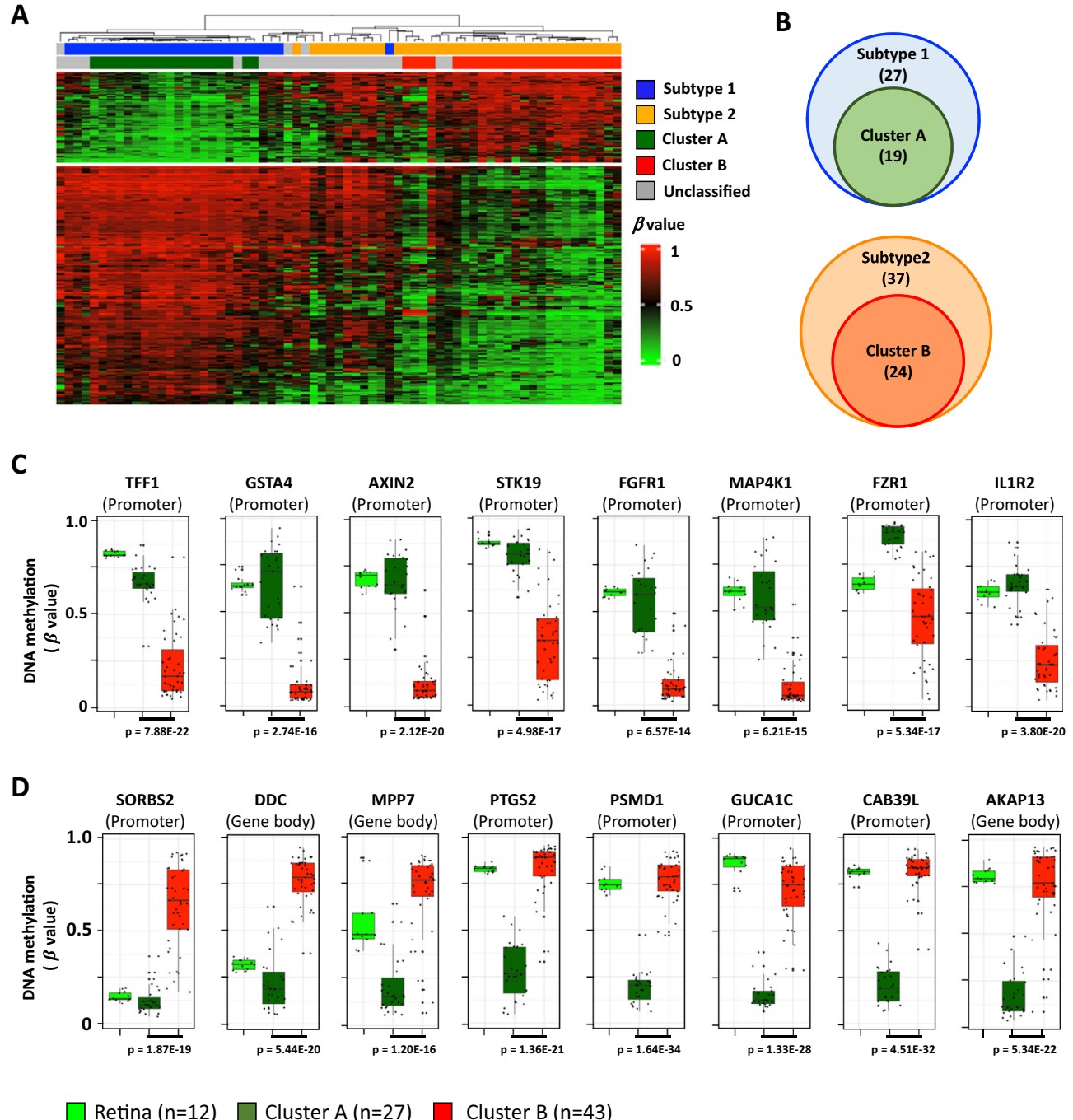

Retina (n=12)    Cluster A (n=27)    Cluster B (n=43)

**Fig. 6 | The DNA methylation signature for RB treatment outcome prediction.**
**A** The selected panel of 320 differentially methylated probes between Clusters A and B in Fig. 5D can separate Subtype 1 (blue) and Subtype 2 (orange) RB samples from the GSE58783 cohort by Liu et al.[36]. By comparison, Clusters A and B are identified by green and red bars, respectively. **B** Venn diagrams showing the overlap of Cluster A (green) with Subtype 1 tumors (blue) and Cluster B (red) with Subtype 2 (orange) tumors. **C, D** Boxplots showing candidate gene DNA hypo- (**C**) and hyper-methylation (**D**) in Cluster B (red) RBs compared to Cluster A (dark green) tumors. The DNA methylation status of each gene in normal retina is shown in light green. For each box plot, the line inside the box denotes the median value, the box denotes the Q3 (top boundary) and Q1 (bottom boundary), and the whiskers denotes the maximum (top) and minimum (bottom) respectively. The *P* value was calculated using two-sided Welch's *t*-tests. Source data are provided as a Source Data file.

tumors are shown in Fig. 6D. Unexpectedly, by including normal retina DNA methylation as a control (bright green), we determined that the majority (7/8) of the hypermethylated genes in Cluster A are similarly hyper methylated in normal retina, while these genes remain unmethylated in Cluster B tumors (Fig. 6C). However, the *FZR1* promoter displayed similar and moderate DNA methylation levels in normal retina and Cluster B tumors, but DNA hypermethylation in Cluster A tumors. Concurrently, most (5/8) hypermethylated genes in Cluster B

were similar to the normal retina, while these genes are unmethylated in Cluster A tumors thus it is the hypomethylation that is aberrant (compared to the retina) (Fig. 6D). However, it should be noted that *SORBS2* promoter and *DDC* and *MPP7* gene bodies showed little to moderate DNA methylation in normal retina and Cluster A tumors, but DNA hypermethylation in Cluster B tumors (Fig. 6D).

Most of the genes identified are known to be involved in tumor aggressiveness (Fig. 6C, D) and may directly contribute to an RB

phenotype that is more likely to fail treatment (or require more aggressive intervention to salvage the eye). *TFF1* overexpression is associated with aggressive disease and correlated with dedifferentiation with stemness features and a higher risk of metastasis in RB[36,37], while *GSTA4* overexpression plays a key role for resistance of cisplatin-chemotherapy[38–40]. *AXIN2* expression is driven by *MYC* and over-expressed in multiple human cancers critical to maintain cancer cell aggressiveness via regulation of the beta catenin/wnt pathway[41]. *STK19* is an NRAS-activating kinase and the over-expression of which leads to cancer invasion and is a potential therapeutic target[42]. *FRGR1* is involved in cancer cell proliferation and metastasis[43], and *IL1R2* promotes cancer cell proliferation and invasion and *IL1R2* blockade suppresses tumor progression[44] (Fig. 6C).

*FZR1* has been described as both a tumor suppressor and onco-protein. *FZR1* promoter DNA hypermethylation in Cluster B tumors may correlate with *FZR1* loss that results in increased sensitivity to DNA damage and resistance to chemotherapy[45]. In addition, *SORBS2* and *CAB39L* have been suggested as potential tumor suppressors[46,47], and silencing of these genes by promoter DNA hypermethylation in Cluster B RBs may contribute to tumor aggressiveness (Fig. 6D). Taken together, these genes not only serve as prognostic biomarkers to predict eye salvage, tumor aggressiveness and likely response to treatment, but opens the door to future applications of predictive medicine by facilitating an in vivo evaluation of potential therapeutic targets for patients with RB, particularly those with more aggressive disease (Fig. 6 and Supplementary Data 2).

## Discussion

There exists a significant body of research into the genetic, genomic and epigenomic alterations of RB. However, this research was done on tumor tissue from surgically removed (enucleated) eyes. Due to the discernable risk of tumor dissemination after tumor biopsy[48,49], obtaining RB tissue DNA has been challenging aside from enucleated specimens. Thus, any application of molecular diagnostic or prognostic biomarkers, or use of these biomarkers for personalized medicine, was limited by the lack of tumor tissue at diagnosis or during therapy. Thus, utilization of a liquid biopsy approach may address this concern for RB and other malignancies in which tumor biopsy is not readily accessible.

Research into the AH liquid biopsy has demonstrated that this biofluid is an enriched source of tumor-derived DNA. In our previous work, we have identified a prognostic genomic biomarker in the AH cfDNA, gain of chr6p, which could predict eye salvage better than currently used clinical classification schemes[10,13,14]. However, this molecular prognostication analysis relies on the presence of somatic copy number alterations in Rb genomes, which not all tumors harbor. In addition, based on the previous investigations[6], ~13% of Rb tumors are initiated by *RB1* promoter hypermethylation. The other genes involved in aggressive RB tumorigenesis, and more importantly, whether they differ between more and less aggressive RB phenotypes, remains an area of active investigation[36].

Aberrant DNA methylation is a common event in most malignancies and a reliable tumor marker for diagnosis and prognosis, however most of the defined alterations appear to be passenger events that do not actually lead to gene expression changes[27,50–52]. The ability to identify RB-derived molecular aberrations in cfDNA isolated from the aqueous humor provides an opportunity to characterize genetic and epigenetic features of eye tumors in vivo while RB patients are actively undergoing therapy[12–14].

In this study, we compared the DNA methylation profiles of primary RB specimens, cfDNA from AH of RB patients, and normal retina tissues, which are available in public databanks. Our analyses revealed that DNA methylation profiles from the tumor-derived cfDNA in the AH is representative of RB tumor tissue[9], thus demonstrating the AH is a reliable biofluid for methylation profiling of the tumor. In this subset

we were able to demonstrate a patient with hypermethylation of the promotor, a known mechanism of tumorigenesis. Previously this could only have been identified from tumor tissue in enucleated eyes; however, the work herein demonstrates the ability to detect this from the aqueous humor (alongside methylation signatures of multiple other genes including *SYK, MYCN*, and others). This opens a multi-omics approach to AH analysis, enabling us to characterize the global methylation pattern of RB tumors in vivo at diagnosis and during therapy, thus obviating the need for tumor tissue. Moreover, unlike genetic alterations, *RB1* epigenetic silencing is reversible and may be a therapeutic target of DNA methylation inhibitors. These have not been used for the treatment of RB, but have been used for treatment of several cancer types in clinical trials[28,53].

By integrating DNA methylation and gene expression data from primary RB tumors, 294 genes were identified that are directly regulated by promoter or gene body DNA methylation. The established correlation between DNA methylation and gene expression in these genes suggests that these DNA methylation markers can be used in place of RNA- or protein-based gene expression profiling. For example, the key therapeutic target genes of *RB1, SYK, MYCN, E2Fs* expression status can be predicted by their DNA methylation status. As expected, *RB1* was also identified as the potential top upstream regulator of genes involved in Estrogen-mediated S-phase entry, and down-regulation of downstream of these genes mimic decreased RB1 activity[54,55]. Notably, the expression changes of these downstream genes controlled by DNA methylation may directly alter the estrogen-mediated S-phase entry independent of *RB1* mutation status, suggesting that DNA methylation is an independent driving force for RB tumorigenesis. Interestingly, these genes also are directly regulated by oncogenic regulators, such as ER, Ras, E2F, MYC, NF-kB, and EGR1 signaling.

Potential prognostic methylation markers for tumor aggressiveness were identified from RB eyes in vivo via an AH liquid biopsy taken at diagnosis or during active treatment. This liquid biopsy approach enabled assay of tumor-derived cfDNA in the absence of tumor tissue. Using the AH, we identified a clear differential methylation signature between eyes that were salvaged with therapy (Cluster A) and eyes that failed therapy and were enucleated (Cluster B). This AH methylation signature is highly concordant with previous genomic and epigenetic analyses of RB tumors[36]. While these results need to be validated in a larger multi-center cohort with various treatment schemes, this work builds upon the important work from Liu et al.[36], suggesting molecular subtypes of RB. Our work allows for detection of these subtypes from the AH in vivo. Once validated prospectively, this has potential for direct impact to these young patients with RB by allowing the clinician to understand the state of the tumor, and combined with clinical features, the likelihood of salvage with various therapies.

Although we did not have the ability to perform gene expression profiling to identify genes whose expression is modulated by DNA methylation in current study, *TFF1, GSTA4, AXIN2, IL1R2, STK19*, and *FRGR1* promoter DNA hypomethylation in aggressive RBs (Cluster B) identified in this study may result in gene overexpression, thereby leading to tumor dedifferentiation with stemness features[36], resistance to cisplatin-chemotherapy[39,40], maintained cancer cell aggressiveness by protecting the tumor from oxidation stress and ensuring MYC-driven transcription[41,56], cancer invasion[42,43], and T-cell suppression[44,57]. Furthermore, silencing of *FZR1* due to promoter DNA hypermethylation in Cluster B cases may decrease sensitivity to chemotherapy[45] and suppress antitumor immunity[44,57]. Interestingly, promoter DNA methylation of tumor suppressor genes *SORBS2* and *CAB39L* may also contribute to tumor aggressiveness characteristic of Cluster B cases[46,47].

DNA methylation is a stable epigenetic modification that is routinely assayed by several technologies[58]. Isolating cfDNA from AH is now a well-established procedure[12,59], and therefore can easily be

applied to RB patients in the clinical setting. Characterizing RB-specific DNA methylation markers in AH cfDNA provides a foundation for future applications in the clinical diagnosis and prognostication of RB and as well as potential for precision medicine-based treatment approaches.

Although our analyses identified regions of interest that may help RB patients in the clinic, these findings are limited by small sample size from a single center. A prospective, multi-center collaborative effort with a large sample size of AH cfDNA from salvaged eyes of RB patients is needed to validate these findings regarding Cluster A and B and assess their clinical relevance with multiple therapeutic options. In addition, future studies could evaluate methylation signatures of cones as an improved control over retina[60].

In conclusion, this study characterizes tumor methylation profiles of RB tumors in vivo using AH liquid biopsy, establishes that the AH methylation signature is highly representative of matched RB tumors, and identifies a cohort of differentially methylated genes with significant potential prognostic utility.

## Methods

### Ethics approval and consent to participate
All human subjects research conducted under this retrospective study was reviewed and approved by the institutional review board at the Children's Hospital Los Angeles (CHLA-17-00248) and following written informed consent from all patients' parents. These experimental methods comply with Helsinki Declaration.

### Consent for publication
Written information consent for publication was obtained from the parents of patients at enrollment.

### Sample collection
Tumor and AH specimens were collected from patients with retinoblastoma at Children's Hospital Los Angeles (CHLA). AH collection was performed at diagnosis or at the time of secondary enucleation and at specified clinical intervals throughout therapy; the methods for AH specimen collection and storage have been previously published[10]. No statistical method was used to predetermine the sample size. For all participants, treatments were performed per routine CHLA protocol. Treatment regimens were unique for each child and only some children had disease recurrence or enucleation. Therefore, a range of biosamples (0–10 AH samples) were collected for each child depending on the clinical course, and blood was drawn alongside AH. HCT116 (CCL-247) human colon cancer cell line was purchased from ATCC with ATCC Cell Line Authentication Service. The growth and passages of cell line was under mycoplasma monitoring.

### DNA extraction from AH, blood plasma, primary tumor samples, and cultured cells
cfDNAs were extracted from AH or blood plasma using the QIAgen QIAamp Circulating Nucleic Acid kit (Qiagen, Valencia, CA USA) as described by the manufacturer. Formalin-fixed, paraffin embedded (FFPE) tumor sections were obtained from the CHLA Pathology Laboratory, and FFPE-DNAs were extracted using the QIAgen QIAamp FFPE DNA Extraction Mini kit as recommended by the manufacturer. cfDNA and FFPE-DNA concentrations were measured using the Qubit dsDNA High Sensitivity Assay system (ThermoFisher, Waltham, MA USA). For analytical validation purpose, 1 μg genomic DNA from HCT116 (CCL-247) human colon cancer cells in 100 μl ddH$_2$O was sonicated 200–300 bp fragment sizes that were verified by agarose gel electrophoresis.

### Bisulfite conversion and restoration
cfDNA and FFPE-DNA samples were subject to bisulfite conversion using the Zymo EZ DNA methylation kit (Zymo Research, Irvine, CA

USA) as specified by the manufacturer. AH cfDNA sample input ranged from 1 to 2 ng, while FFPE-DNA sample input ranged from 160 to 240 ng. The amount of bisulfite-converted DNA as well as the completeness of bisulfite conversion for each sample are assessed using a panel of MethyLight-based real-time PCR quality control assays[61]. Bisulfite-converted DNAs are then subjected to the Illumina EPIC BeadArrays, as recommended by the manufacturer and described by Moran et al.[62].

### Targeted bisulfite sequencing
Bisulfite-converted DNA was amplified by PCR using the following primers (5′ to 3′) targeting: (1) *TFF1* promoter (forward: GGG AAA GAG GGA TTT TTT GAA TT, REVERSE: AAC TAC CAA AAC TAA CTA TAA CCC CAC AA), (2) *HOXC4* promoter (*forward*: ATT TAT TTA AGT GTT AAT TAG GTT GGG T; reverse: AAT TTA AAA TCA TAA CTT ACC AAA ACT CAA), (3) *MNX1* gene body (forward: GGG ATT TGA GGG ATA GTG ATT T, REVERSE: CAA AAT TCA AAT TTC AAC CCC CTA A) and (4) *CELSR3* gene body (forward: AGT ATT GGG AGT TAT TTT TGA GGT T, *REVERSE*: CAA TCC TCT CCT AAA AAC CAA A). PCR products were then sequenced using Amplicon-EZ service (Genewiz). Sequencing data were analyzed using the EPIC TABSAT tool[63].

### EPIC DNA methylation data production
DNA methylation was evaluated using the Illumina Infinium MethylationEPIC (EPIC) BeadArray at the USC Norris Molecular Genomics Core Facility. Specifically, each bisulfite converted sample was whole genome amplified (WGA) and then enzymatically fragmented. Samples were then hybridized overnight to an 8-sample EPIC BeadArray, in which the amplified DNA molecules anneal to locus-specific DNA oligomers linked to individual bead types. The oligomer probe designs follow the Infinium I and II chemistries, in which cy3/cy5-labeled nucleotide base extension follows hybridization to a locus-specific oligomer. After the chemistry steps, BeadArrays were scanned using the Illumina iScan system to generate the *.IDAT files in both red and green channels. Raw signal intensities were extracted from the *.IDAT files using the 'noob' function in the *minfi* 1.40.1 R package and the B5 version of the probe manifest. The 'noob' function corrects for background fluorescence intensities and red-green dye-bias developedby Triche et al.[21]. The beta (β) value represents the DNA methylation score for each data point and is calculated as (M/(M + U)), in which M and U refer to the mean methylated and unmethylated probe signal intensities, respectively. β values range from 0 to 1, with zero indicating an unmethylated locus and one indicating a fully methylated locus. Measurements in which the fluorescent intensity is not statistically significantly above background signal (detection $p$ value > 0.05) as well as non-specific probes and those on the X- and Y-chromosomes were removed from the data set.

### Data analysis
DNA methylation data of normal retina and RB tumors were obtained from the Gene Expression Omnibus (GEO, GSE57362 and GSE58783)[9,36]. Sample purity was assessed using the LUMP (leukocytes unmethylation for purity) assay[64] and 27 samples with LUMP values <0.5 (<50% purity) were removed from further analysis (Fig. S2). Probes related to gender and age, and as well as those overlapping known polymorphisms were also excluded from further analysis[20,65]. Differentially methylated probes were selected using absolute mean β-value difference > 0.3 between normal retina and RB tumor samples from GSE58783. Two-sided Welch's *t*-test (R package *matrixTests*) was used to identify statistical significance ($p$ value < 0.05). The top 10,000 probes with the greatest β-value standard deviation (SD) across the four pairs of matched RB primary tumors and AH samples (RB_CHLA_1–4 and AH_CHLA_1–4) were also selected for DNA methylation comparisons. Heatmaps were generated using the R package *ComplexHeatmap*[66], and multidimensional scaling

(MDS) plots were generated using 'plotMDS' function in the *edgeR* package[67].

## Probe annotations
Probe annotations were obtained from the B5 version of the Infinium MethylationEPIC probe manifest (hg19, illumina.com). We defined "Promoter" probes as those located at the transcription start site (TSS200 or TSS1500), 5' untranslated regions (UTR) and the first exon. In addition, we classified "Gene Body" probes as those located within gene bodies and 3'UTRs. The remaining probes were classified as "Other" probes as those not included in Promoter or Gene Body categories.

## Differentially expressed genes
Gene expression array data (GSE125903 and GSE111168)[29,30] were used to identify differentially expressed genes between apparently normal retina and RB tumors. The processed expression data were downloaded from BaseSpace correlation engine[68]. The upregulated genes and down-regulated genes (fold change > 2 or <−2) overlapped from both datasets were used for further analysis.

## Pathway and TF binding motif analyses
Pathway analyses for genes regulated by DNA methylation were performed using Ingenuity Pathway Analysis (QIAGEN) (https://digitalinsights.qiagen.com/products-overview/discovery-insights-portfolio/analysis-and-visualization/qiagen-ipa/). The transcription factor (TF) binding motif prediction was performed using F-match analysis from the TRANSFAC 2.0 database (genexplain, Germany)[32].

## Copy number variation analysis
Copy number variation was detected using the R package *SeSAMe*[21,69]. The stored normal data (EPIC.5.SigDF.normal) from sesameData was used for normalization.

## Reporting summary
Further information on research design is available in the Nature Research Reporting Summary linked to this article.

# Data availability
The raw DNA methylation datasets generated for this study are publicly available at the Gene Expression Omnibus GSE208055 and GSE211508. TRANSFAC (2.0, genexplain, Germany) [https://genexplain.com/transfac] was used for transcription factor binding sites prediction. The previously published public DNA methylation data of normal retina and RB tumors used in this study are available in GEO database under accession codes GSE57362 and GSE58783. The previously published, processed expression array data (GSE125903 and GSE111168) were downloaded from BaseSpace correlation engine. The remaining data are available within the Article, Supplementary Information, or Source Data File. Source data are provided with this paper.

# Code availability
The codes used to generate for the analysis, figures in this study are available at Github repositories (https://doi.org/10.5281/zenodo.7005924)[70]. R(4.1.1), Rstudio (1.4.1106), and R packages (ggplot2 3.3.5, stats 4.1.1, matrixTests 0.1.9.1, and circlize 0.4.1.4) were used in this study.

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

## Acknowledgements

We thank the retinoblastoma cancer patients and their families for their participation and for providing specimens to the advancement of cancer research. We are grateful to the research support provided by CHLA Team RAPIDO: Brianne Brown, Mark Reid, Dilshad Contractor & Armine Begijanmasihi. We thank the Norris Translational Pathology Core & the Pediatric Research Biorepository for preparing the specimens for this study, and the USC Molecular Genomics Core. This study is supported by the Vicky Joseph Cancer Research Foundation (G.L.), the National Institute of Health (R35 CA209859) (G.L.), National Cancer Institute (P30 CA014089) (G.L., D.J.W., and J.L.B.). National Cancer Institute of the National Institute of Health (K08CA232344) (J.L.B.), 2R01CA137124 (D.C.), National Institute of Health (P30EY029220) (J.L.B.), Hyundai Hope on Wheels, Wright Foundation, The Knights Templar Eye Foundation, The Berle & Lucy Adams Chair in Cancer Research, The Larry and Celia Moh Foundation, The Institute for Families, Inc., Children's Hospital Los Angeles, and an unrestricted departmental grant from Research to Prevent Blindness.

## Author contributions

L.X. and J.L.B. conceptualized the project. H.T.L., L.X., G.L, D.C., and J.L.B. curated collected data. H.L., D.J.W., M.L., W.Z., C.C.P., and K.S. performed data analysis. H.L., D.J.W., M.L., W.Z., and G.L. performed methodological developments. L.X., G.L., and J.L.B. oversaw project administration. H.L., M.L., W.Z., and G.L. developed analysis software algorithms. D.C., G.L., and J.L.B. supervised studies. H.L. and L.X. visualized data and analysis. H.L., L.X., and G.L. performed writing of original manuscript draft. All authors contributed to the investigation including performed writing, review, preparation, and editing of the manuscript.

## Competing interests

The authors declare no competing interests.
