## [Peer Review File · Nature Communications]

REVIEWER COMMENTS

Reviewer #1 (Remarks to the Author): expertise in retinoblastoma from a clinical perspective

These results are noteworthy and of potential high significance to the care of children with RB in the future. The strong importance of methylation control in this special cancer, RB, that has a history of leading cancer in general in genetics, make this work significant and of future potential importance for other cancers.

The use of cfDNA make it easy to develop a prospective clinical trial to really test the importance of the AHsignal at diagnosis to guide treatment options and potential to avoid needless dangerous therapy when primary surgery would be curative.

Will the work be of significance to the field and lated fields? YES

How does it compare to the established literature? NO PREVIOUS LITERATURE this is entirely original.

Does the work support the conclusions and claims, or is additional evidence needed? The claims are overstretched given the very small "n" in the real AH samples, analyzed extensively.

Are there any flaws in the data analysis, interpretation NO

BUT conclusions are overstated.

Do these prohibit publication NO

YES to revision YES and the paper would be more meaningful.

NO change requires language but NO CHANGES to data or figures except 1 minor inconsistent label.

Is the methodology sound? YES

Does the work meet the expected standards in your field? YES except for over-interpretation as mentioned above.

Is there enough detail provided in the methods for the work to be reproduced? YES

Reviewer #2 (Remarks to the Author): expertise in DNA methylation

The manuscript by Li and colleagues describes an interesting new way of utilizing cell free DNA methylation to understand retinoblastoma tumor etiology and inform biomarker development. Rather than use plasma this group used the aqueous humor of the eye to sample the primary tumor since it is not clinically advised to directly biopsy this type of tumor. In a small set of Rb patients they use the 850k array to compare AH DNA methylation to matched tumor DNA methylation, and showed good concordance between the tumor and AH DNA were available from the same patient. They then mine their DNA methylation data for genes that may be involved in driving Rb by integrating their 5mC data with public tumor RNA seq. Next, the authors mine their AH data for 5mC signatures of favorable outcome tumors (successful salvage therapy) vs those more aggressive tumors that required enucleation. Again, the sample size is small, but this approach identified a compelling signature that segregates primary RB tumors and, more interestingly, this signature from salvage vs enucleated eyes based on AH methylation correlated very strongly with an independently derived signature based on 5mC, CNV and several other parameters that divided Rb tumors into good outcome differentiated vs poor outcome undifferentiated tumors. Overall, the primary weaknesses of the study, as the authors note, is the small sample size, as well as lack of confirmatory or functional data. However, the study scores very highly in significance and impact and as such I am recommending a high level of enthusiasm for the study with relatively minor revisions only.

Issues to be considered:

Small sample size. It would be fruitless to request a much larger cohort of AH samples and/or a replication set since this is a rare tumor and relatively new methods. But the authors should nonetheless consider whether any additional samples and/or any confirmatory type of study could be added. With regard to the later this could include analysis of expression/methylation (pyrosequencing/RT-PCR) for a gene of interest in primary Rb tumor tissue or even an RB cell line if they exist (for example a gene they highlight like FZR1).

Enhancers are noted in fig 1C, how are these defined? Is this a generic enhancer annotation or is data available to create an 'eye' specific set of enhancers, then interrogate these for methylation changes.

Use of the term 'driver genes' is misleading as this term typically is associated with some functional readout like growth promotion or EMT.

Why not call CNV from the data and integrate that with both their previous data and public sources like the Liu et al. study? Seems like a lost opportunity.

A number of genes of interest are described (cyclins, p16, FZR1...), are these methylation targets enriched for E2F binding sites in their promoters? A more general analysis of TF binding associations in the methylation, regulated genes could prove illuminating.

230 differentially methylated probes (p. 13) between clusters A and B – how many genes does this correspond to, pathway enrichments on this key gene set are needed, and they should be provided as a gene list in a suppl table.

Dendrograms (trees) are missing from many of the heatmaps and should be added back. This is especially true for the heatmaps in figs. 3, 5, and 6. Without them it makes it almost impossible to appreciate any groups within the samples. Related, the PCAs in fig. 5 are basically unreadable. They should either be made larger or perhaps put in the supplement (so they can be much larger).

Reviewer #3 (Remarks to the Author): expertise in epigenetics of retinoblastoma

In their manuscript titled "Characterizing DNA methylation signatures of retinoblastoma using the aqueous humor liquid biopsy: moving beyond genomics," Li et al. perform DNA methylation array analysis (450k) on 4 primary tumors and 15 samples of aqueous humor and 2 blood samples. They compare their data to some previously published data. They also performed some quality control assays to determine the limit of detection for the arrays and tried to use their analyses to predict molecular and prognostic features of retinoblastoma such as RB promoter hypomethylation and outcome. Overall, this manuscript is a small study with a small number of samples and there are several previously published studies on DNA methylation in retinoblastoma. In particular, sampling of aqueous humor for DNA analyses has been previously published in several papers and there is at least 1 review article on the topic. Thus, this study is too small to advance the field and the prognostic significance has not been demonstrated in a large cohort of patients. More importantly, this study assumes there are cell intrinsic features of retinoblastoma that correlate with outcome as for other tumors. While this may be the case for some tumors, there are several other features that may not have anything to do with the intrinsic properties of the tumor. For example, location within the eye may be stochastic yet this could have a major impact on local and distal cell invasion and metastasis. Vitreal seeds may not be an intrinsic property but again, this is an important clinical feature. Also, there is no standard for when to enucleate an eye in certain subsets of patients and when to attempt therapy. For example, the group at MSKCC may have a very different approach to treatment than other centers who don't specialize in RB or take such an aggressive treatment approach.

Response to Reviewers

We are pleased to resubmit a revised version of NCOMMS-22-11089-T for publication and truly appreciate the constructive criticisms made by the reviewers. We have incorporated the suggested edits into the revised manuscript, including updated Figures 3, 4, and 5, as well as Supplemental Figure S1, and the addition of Supplemental Figure S3, S5 and S6, to address each of the reviewer's concerns as outlined below.

First, we want to thank the reviewers for their positive comments and suggestions. The reviewers' concerns focused on both notational and broader inquiries related to the study's findings. Following the reviewers' advice, we have fixed the errors in the manuscript, revised figures, and added comments and clarifications to the manuscript text (see specifics outlined below). All changes in the text are highlighted in red – we hope that these will greatly strengthen the manuscript for publication in Nature Communications.

Reviewer #1

(Remarks to the Author): expertise in retinoblastoma from a clinical perspective

These results are noteworthy and of potential high significance to the care of children with RB in the future. The strong importance of methylation control in this special cancer, RB, that has a history of leading cancer in general in genetics, make this work significant and of future potential importance for other cancers. The use of cfDNA make it easy to develop a prospective clinical trial to really test the importance of the AH signal at diagnosis to guide treatment options and potential to avoid needless dangerous therapy when primary surgery would be curative.

Response: *We appreciate the reviewer's supportive comments.*

Li et al present convincing data that methylation signatures in Retinoblastoma (RB) tumors distinguish significantly different pattern from normal retina and variable patterns in RB tumor. Previous data of some of these authors showed that cfDNA from the anterior chamber of eyes that are retained (*in vivo*), accurately represents genomic changes in *RB1* and other genes modified in enucleated tumors. Now they analyse the methylome of enucleated tumors and cfDNA from the eye anterior chamber. They show different methylome of genes broadly known to be important in cancer evolution, including those in the pathway of action of *RB1*.

Comments: Overall, extra words often cloud the meaning of sentences:

Abstract. *"We identified 294 genes that are directly regulated by DNA methylation are enriched in RB and p53 tumor suppressor (RB1, p53, p21, and p16) and oncogenic pathways (E2F)."* Does "enriched" mean that, in RB tumors, DNA methylation is activated/suppressed in genes that are known to function in tumor suppressor and oncogenic pathways?

Response: *Yes, this is correct. We have simplified the wording in the revised manuscript page 2 to read: We identified 294 genes directly regulated by DNA methylation in RB that*

are implicated in p53 tumor suppressor (RB1, p53, p21, and p16) and oncogenic pathways (E2F).

This study identifies markers in AH that are prognostic for eye salvage vs failure for retinoblastoma. How is ??? aggressiveness defined from eyes that have not been enucleated?

Response: *Agreed. We have changed this text to read: Finally, we identified AH DNA methylation prognostic signatures for eye salvage versus treatment failure (enucleation) for retinoblastoma patients. Please find this in the revised Abstract in page 2.*

Introduction:

This could be shortened to 2 paragraphs with removal of extra, redundant words, etc. and then precisely offer some definitions for the reader to more clearly read the rest of paper.

Response: *Yes, we have cleaned up the text in the Introduction section of the revised manuscript.*

“In addition, epigenetic deregulation of tumor-promoting pathways has been shown to be important for further RB tumorigenesis and disease progression”⁷⁻⁹. “What is “further RB tumorigenesis...”? Does that mean failure of attempted salvage? Or Group E eyes? (the classification of the salvaged eyes may be in a Supplementary Table but otherwise not mentioned)

How does the later use of the word “aggressive” relate to this? *“.....biomarker of aggressive disease more likely to fail salvage therapy”^{12,13}.*

Response: *We apologize for the confusion; we mean tumorigenesis beyond the initial RB1 loss (which is thought to initiate a retinoblastoma and further molecular events are required for retinoblastoma formation. The corrected sentence is: “In addition, epigenetic deregulation of tumor-promoting pathways has been shown to be important for RB tumorigenesis and disease progression beyond RB1 inactivation” in page 2 of the revised manuscript.*

In terms of aggressive, we have clarified this in page 3 to read: as well as DNA methylation profiles that may predict an aggressive tumor subtype less likely to respond to medical therapy.

Results:

Where is the data on aggressive disease? I think there are 34 primary enucleations, maybe because of disease severity?

Response: *We are evaluating the definition of aggressive disease by Liu et al³² (subtype 2) which are tumors associated with dedifferentiation, stemness features and expression of neuronal markers; this requires a tumor biopsy which is not available outside of enucleation. Thus, in the clinical context we can evaluate aggressive tumor subtypes as*

those less likely to respond to medical therapy and be cured. This is best done with the (admittedly small) CHLA cohort of CHLA_SV vs. CHLA_PE, SE.

Are all GSE57362 sample from primary enucleation?

Response: *Unfortunately, the PE or SE status of these surgically removed tumors is not available.*

The 4 RB_CHLA primary enucleations will have available clinical data, but I suspect that will be thin for the 30 RB_SR-1-30 samples studied.

Response: *Yes, that is unfortunately correct, but it is the publicly available data we have. The clinical information for the CHLA group is listed in Table 1.*

Terminology is inconsistent: “secondary enucleation (ES)” to be consistent with “PE”, should be **SE**

Response: *We apologize for the inconsistency and have replaced ES with SE throughout the revised manuscript.*

Fig 5: A very small sample sizes to define aggressive. Yes, data in 5A is very clear, but the small “n” is a weakness of the overall predictive conclusion, pointing to a study weakness and to need for clinical trial with AH sampling of patients at diagnosis then follow standard of care and outcomes.

Response: *We agree! And are aware of the small sample size and the need to validate our findings. This challenge is compounded as RB is a rare cancer and sample availability is limited, especially for AH cfDNA specimens. However, our prognostic signatures (Cluster A and B) were validated in two datasets involving over n=70 primary tumor specimens. In addition, we have validated the prognostic signatures that determine Subtype I and II RBs. Currently, we are collecting AH cfDNA and primary RB specimens for additional future validation experiments. We are also establishing RB cell lines for which we will interrogate the potential roles of the genes regulated directly by DNA methylation as described in Figures 4 and 6.*

We have added the following statement to the results in page 18: While these results need to be validated in a larger cohort and with various treatment schemes; this work builds upon the important work from Liu’s recent work³². suggesting a more and less aggressive molecular subtype of RB – our work allows for detection of this from the AH in vivo. Once validated prospectively, this has potential for direct impact to these young patients with RB by allowing the clinician to understand the state of the tumor and the likelihood of salvage with various therapies.

Fig 1 primary rb vs retina

Fig S1 could be more useful if the nomenclature and color labelling of the figures was included.

Response: *We thank the reviewer for this suggestion and have updated **Supplemental Figure S1** to include a table.*

Fig S2 has incorrect annotation for CHLA samples (ie should be RB_CHLA.....etc)

Response: *Thank you for this comment. We apologize for the error. This is corrected in the revised Fig S2 legend.*

Figure 3C:

Is the hypermethylated promoter in sampled AH_CHLA_3 and RB_CHLA_3, (same patient) and RB_SR_24 and RB_SR_30 evident as *RB1* loss of heterozygosity and homozygous *RB1* promoter methylation in standard genetic testing?

Response: *Great question! Standard clinical testing for RB is done on peripheral blood samples to identify germline *RB1* pathogenic variants that predispose to the development of RB and other secondary tumors. Hypermethylation of the *RB1* promotor, and well-known cause of non-germline, sporadic RB, cannot be identified via this mechanism (eg via the blood) – previously this could only be identified with access to tumor tissue however can now be identified using the aqueous humor (here’s a good ref: <https://pubmed.ncbi.nlm.nih.gov/29851531/>).*

Figure 5:

The major conclusion of the paper is based on 12 AH samples from CHLA and the **real** comparison, I think, is between 4 AH_SV (salvaged) and 4 AH_ES (secondary enucleation, therefore assumed “aggressive” tumor).

The very small sample sizes are used to define aggressive. Yes, data in 5A is very clear, but the small “n” is a weakness of the overall predictive conclusion, pointing to a study weakness and to **need for clinical trial with AH sampling of patients at diagnosis then follow standard of care and outcomes**. the conclusions would be stronger to state that this data has potential to guide selection of eyes to attempt salvage, but this paper does not show this to be better than the standard of care based on clinical features defining Group E/ cT3 stage disease.

Response: *Agreed! And we are working towards that exact goal. Our laboratory has shown that identification of 6p gain in the AH performs better than Group classification for predicting eye salvage however we need a similar prospective approach to the addition of these epigenetic biomarkers/signatures.*

As per above we have added the following clarifying statement in page 18 about need for prospective validation of these findings: While these results need to be validated in a larger cohort and with various treatment schemes; this work builds upon the important work from Liu’s recent work³² suggesting a more and less aggressive molecular subtype of RB – our work allows for detection of this from the AH in vivo. Once validated prospectively, this has potential for direct impact to these young patients with RB by

allowing the clinician to understand the state of the tumor and the likelihood of salvage with various therapies.

This table helped to understand the paper and would benefit other readers (improved of course from my home-made attempt).

Response: *Thank you! We appreciate this suggestion and have included this table in Supplemental Figure 1.*

Reviewer #2

(Remarks to the Author): expertise in DNA methylation

The manuscript by Li and colleagues describes an interesting new way of utilizing cell free DNA methylation to understand retinoblastoma tumor etiology and inform biomarker development. Rather than use plasma this group used the aqueous humor of the eye to sample the primary tumor since it is not clinically advised to directly biopsy this type of tumor. In a small set of Rb patients they use the 850k array to compare AH DNA methylation to matched tumor DNA methylation and showed good concordance between the tumor and AH DNA were available from the same patient. They then mine their DNA methylation data for genes that may be involved in driving Rb by integrating their 5mC data with public tumor RNA seq. Next, the authors mine their AH data for 5mC signatures of favorable outcome tumors (successful salvage therapy) vs those more aggressive tumors that required enucleation. Again, the sample size is small, but this approach identified a compelling signature that segregates primary RB tumors and, more interestingly, this signature from salvage vs enucleated eyes based on AH methylation correlated very strongly with an independently derived signature based on 5mC, CNV and several other parameters that divided Rb tumors into good outcome differentiated vs poor outcome undifferentiated tumors. Overall, the primary weaknesses of the study, as the authors note, is the small sample size, as well as lack of confirmatory or functional data. However, the study scores very highly in significance and impact and as such I am recommending a high level of enthusiasm for the study with relatively minor revisions only.

Response: *Thank you! We appreciate the reviewer's positive comments and are also enthusiastic about the potential to better care for children with RB once this work is appropriately validated, prospectively.*

Issues to be considered:

Small sample size. It would be fruitless to request a much larger cohort of AH samples and/or a replication set since this is a rare tumor and relatively new methods. But the authors should nonetheless consider whether any additional samples and/or any confirmatory type of study could be added. With regard to the latter this could include analysis of expression/methylation (pyrosequencing/RT-PCR) for a gene of interest in primary Rb tumor tissue or even an RB cell line if they exist (for example a gene they highlight like *FZR1*).

Response: *We agree and thank the reviewer for understanding some inherent limitations in managing a rare tumor. The small sample size was also mentioned by Reviewer 1. We are aware of the small sample size for discovery and validation. However, because RB is a rare cancer, available samples, most notably AH cfDNAs, are limited however collection of samples is active and ongoing. Illumina-based DNA methylation array data are reliable, and we have previously confirmed DNA methylation data using pyrosequencing of urine sediment DNA for bladder cancer (Su et al, Clinical Cancer Research 2014) or targeted bisulfite sequencing for primary kidney tumors (Becket et al, Cancer Research, 2016). In this study, we use targeted bisulfite sequencing to validate the EPIC DNA methylation array data as the reviewer suggested (**Supplemental Figure S5**).*

We performed candidate gene bisulfite sequencing of four gene regions on 10 primary RB tumors and one apparently healthy retina. Please note these genes were selected based on following features: 1) dramatical methylation different between retina and RB tumors and directly regulated by DNA methylation in Figure 4 and supplemental table 1; 2) CpG rich or CpG island; 3) allowing us to do target bisulfite sequencing which need have two primers without CpG site. The four gene regions are at the TFF1 promoter, HOXC4 promoter, MNX1 gene body and CELSR3 gene body. Comparing these data to the EPIC DNA methylation array data for these same samples showed high correlation for all four loci. Unfortunately, we do not have RNA for these specimens to confirm the correlation between DNA methylation and gene expression for these four genes. However, the correlation of DNA methylation (DNA methylation array) and gene expression (RNA-seq) for TFF1 has been confirmed by Liu et al³².

Currently, we are not only collecting AH cfDNA and surgical tumor specimens for future validation, but also establishing RB cell lines to study these novel genes regulated by DNA methylation that are presented in Figures 4 and 6.

Enhancers are noted in fig 1C, how are these defined? Is this a generic enhancer annotation or is data available to create an 'eye' specific set of enhancers, then interrogate these for methylation changes.

Response: *The potential enhancers were identified by Illumina are based on available databases for potential enhancers for any type of cells or tissues. However, these are not specific for retinal cells (Pidsley et al, Genome Bio. 2016; Zhou et al, NAR, 2017).*

Use of the term 'driver genes' is misleading as this term typically is associated with some functional readout like growth promotion or EMT.

Response: *We have changed this term to "potential novel or aggressive genes regulated by DNA methylation" in the revised version of the manuscript.*

Why not call CNV from the data and integrate that with both their previous data and public sources like the Liu et al. study? Seems like a lost opportunity.

Response: *We agree, however, DNA methylation data from public databases are missing some files required for this analysis. For example, the deposited methylation array data are in a txt file format rather than the .idat file format. In addition, the out-of-band signals are missing, thus leading to technical difficulties in analyze those data. We analyzed our own EPIC DNA methylation array dataset for four pairs of primary tumors and AH cfDNAs and found high correlation between primary tumors and AH cfDNA for both DNA methylation (Figure 2C) and CNV (**Supplemental Figure S3A**). In addition, fewer CNVs were observed in salvaged samples using EPIC array data, and we included the analyses of Supplemental Figure S3 in this revised manuscript. Indeed, we included the following statement: “As expected, based on our previous work and that of Liu et al³², salvaged eyes had fewer copy number alterations than enucleated eyes, especially for gain of 1q, 6p and loss of 16q (**Supplemental Figure S3B**)” on page 11.*

A number of genes of interest are described (cyclins, p16, FZR1...), are these methylation targets enriched for E2F binding sites in their promoters? A more general analysis of TF binding associations in the methylation, regulated genes could prove illuminating.

Response: *We thank the reviewer for this suggestion. We performed TRANSFAC analysis to identify overrepresented transcription factor binding sites for DNA methylation regulated genes and RB-specific differentially expressed genes (**Supplemental Figure S6**). The top over-represented transcripts that target the promoters of DNA methylation-regulated genes in Figure 4B include oncogenic regulators involved in ER (Estrogen receptor), Ras (RREB-1, Ras responsive element binding protein 1), E2F, MYC (MAZ, MYC associated zinc finger protein), NF-kB, and EGR1 (Early growth response protein 1) signaling (Supplemental Figure S6).*

230 differentially methylated probes (p. 13) between clusters A and B – how many genes does this correspond to, pathway enrichments on this key gene set are needed, and they should be provided as a gene list in a suppl table.

Response: *The set of 320 probes corresponds to 185 unique genes. We have added this information to **Supplemental Table S2**.*

Dendrograms (trees) are missing from many of the heatmaps and should be added back. This is especially true for the heatmaps in figs. 3, 5, and 6. Without them it makes it almost impossible to appreciate any groups within the samples. Related, the PCAs in fig. 5 are basically unreadable. They should either be made larger or perhaps put in the supplement (so they can be much larger).

Response: *We thank the reviewer for this suggestion. We have updated the heatmaps in Figures 3, 5 and 6 to include the heatmap dendrograms.*

Reviewer #3 (Remarks to the Author): expertise in epigenetics of retinoblastoma In their manuscript titled "Characterizing DNA methylation signatures of retinoblastoma using the aqueous humor liquid biopsy: moving beyond genomics," Li et al. perform DNA methylation array analysis (450k) on 4 primary tumors and 15 samples of aqueous humor

and 2 blood samples. They compare their data to some previously published data. They also performed some quality control assays to determine the limit of detection for the arrays and tried to use their analyses to predict molecular and prognostic features of retinoblastoma such as RB promoter hypomethylation and outcome. Overall, this manuscript is a small study with a small number of samples and there are several previously published studies on DNA methylation in retinoblastoma. In particular, sampling of aqueous humor for DNA analyses has been previously published in several papers and there is at least 1 review article on the topic. Thus, this study is too small to advance the field and the prognostic significance has not been demonstrated in a large cohort of patients.

Response: *We thank the reviewer for their time and review of our work. We agree that this is a small sample sizes, limited by a rare disease with few samples of AH taken at diagnosis. We have addressed the need to prospectively validate this work due to the small sample size.*

While there have been multiple studies on DNA methylation in RB, these were done on tumor tissue only. Until recently, there was a complete lack of molecular information from salvaged eyes due to the contraindication to biopsy this tumor. Now, with access to AH cfDNAs, including from salvaged eyes, there is finally potential to put decades of research on RB tumor tissue into a clinically impactful assay for patients in the absence of tumor tissue – the AH can be assayed at diagnosis and throughout therapy – and used for molecular assay. Our laboratory has pioneered the AH liquid biopsy with multiple publications, however none of them address epigenetic analysis via the AH. Other laboratories have validated our work but again, none address methylation signatures in vivo via the AH.

There are multiple clinically impactful applications of this work – once validated. For example, the simple ability to identify hypermethylation of the RB1 promoter via the AH (as shown with AH_CHLA_3) in the absence of tumor tissue is exciting and impactful. This alteration is sporadic, and non-heritable which aids in genetic counselling. In the future it may even be possible to restore function of the gene in these patients.

This entire field of AH liquid biopsy allows us to finally evaluate the molecular profiles of the tumors in each eye to better understand how these molecular alterations help explain clinical behavior of these tumors, including poor response to medical therapy. In fact, one of our more recent papers discusses the distinct molecular alterations between two eyes from the same patient which possibly explains why one eye responded well to therapy, while the other eye did not (Wong et al NPJ Precis Oncol 2021 <https://pubmed.ncbi.nlm.nih.gov/34316014/>).

More importantly, this study assumes there are cell intrinsic features of retinoblastoma that correlate with outcome as for other tumors. While this may be the case for some tumors, there are several other features that may not have anything to do with the intrinsic properties of the tumor. For example, location within the eye may be stochastic yet this could have a major impact on local and distal cell invasion and metastasis. Vitreal seeds

may not be an intrinsic property but again, this is an important clinical feature. Also, there is no standard for when to enucleate an eye in certain subsets of patients and when to attempt therapy. For example, the group at MSKCC may have a very different approach to treatment than other centers who don't specialize in RB or take such an aggressive treatment approach.

Response: *This is a fantastic comment, and we agree. All of the clinical features listed here (tumor size, seeding, distant seeding, location) are part of the clinical Grouping scheme currently used to prognosticate the likelihood of eye salvage for this tumor. Our group has already shown that genomics outperforms clinical classification schemes in predicting eye salvage (Berry et al Mol Cancer Res). And in fact, this has been true also for other tumors and even other ocular tumors such as uveal melanoma wherein gene expression profiling of the tumors is better able to predict clinical outcomes than clinical features alone (including size, location, etc).*

Of course, what we have seen with other tumors, is that both clinical features and molecular features matter – and we suspect the same will be true for RB. For example, a large Group D tumor with seeding with an ‘aggressive methylation signature’ is likely to respond worse to therapy than a smaller Group B tumor with the same profile.

To better understand that requires prospective validation, which we acknowledge in the manuscript will need to be done before any clinical application of this work can take place.

However, what cannot be underestimated is the need for molecular prognostication for this tumor at diagnosis to help guide therapy. The reviewer references an aggressive treatment approach at MSKCC – maybe that’s not needed for every child and some could be spared aggressive treatment. Maybe the aggressive phenotype requires 3 chemotherapeutic agents and the less aggressive subtype only required 1. All of this is new, uncharted therapy due to advances in liquid biopsy. As a clinician who treats children with RB at a large referral center, I can say with certainty that it would be a game-changer for this disease to have an objective test that could help guide therapy and the likelihood of responding (thus making a more informed risk-benefit analysis for the parents).

Finally, it is even possible that this work will open the door for precision therapies aimed at what is driving the tumor at the cellular level. There are a lot of questions that require further investigation and research. What is exciting and novel is that we can now assay this information using the AH platform in vivo without tumor tissue – so that any future validated assay that builds from the work presented herein will actually be applicable to the child and their treatment regimen.

Our results suggest that AH cfDNAs have potential utility in diagnostic and prognostic purposes including identification of potential therapeutic targets for RB clinical management. Further detailed in vitro and in vivo studies are needed to confirm the mechanisms of epigenetic regulation and roles of the novel genes we have identified in this study.

REVIEWERS' COMMENTS

Reviewer #1 (Remarks to the Author):

The reviewers' queries have been very well addressed.

Minor corrections:

page 17 lines 401-404

"While these results need to be validated in a larger cohort and with various treatment schemes; this work builds upon the important work from Liu's recent work . suggesting a more and less aggressive molecular subtype of RB – our work allows for detection of this from the AH in vivo."

Grammar is not correct, and ref to "Liu's recent work" to be included, as follows:

"While these results need to be validated in a larger cohort and with various treatment schemes, this work builds upon the important work from Liu's recent work REF, suggesting a more and less aggressive molecular subtype of RB. Our work allows for detection of subtype from the AH in vivo."

Reviewer #2 (Remarks to the Author):

The authors have done an excellent job at addressing prior concerns. No additional concerns are noted. The study is novel and high impact, and in time could lead to improved care for RB patients based on AH cfCNA and epigenetic signatures. I recommend the paper be accepted.

Reviewer #3 (Remarks to the Author):

While I agree that AH DNA methylation analysis may one day complement the other features of retinoblastoma that are used for staging, it is unlikely that it will ever stand alone because of the complexity of the disease. As noted, location of the primary tumor (near optic nerve, macula, vitreal seeds, anterior chamber), possibility of preserving vision and family preference for preserving the eye in the absence of vision are all important considerations. My major concern with this manuscript are:

1) feasibility of AH DNA methylation has been demonstrated and the next step in the field is a large prospective study through a cooperative group. This tiny study does not advance the field.

2) Claims that the underlying molecular mechanism can be inferred from DNA methylation are not supported by the data. While a small percentage of tumors may have hemizygous promoter hypermethylation, there is no treatment specific to those patients. Moreover, there are no druggable genetic or epigenetic targets in retinoblastoma despite extensive studies.

Therefore, the two major claims of the paper are not supported by the data. They have not demonstrated that DNA methylation can be used to stratify patients for treatment (study is much too small to change practice). They have not demonstrated any evidence that the DNA methylation data can be used to customize treatment.

Response to Reviewers

We are pleased to resubmit for publication a revised version of this manuscript. We thank the reviewers for their positive comments and suggestions. Following the reviewers' advice, we have fixed the errors in the manuscript, revised certain figures, added comments and clarifications to the manuscript text (see specifics outlined below), and highlighted all changes as red in the text, all of which we hope will greatly strengthen the submission. We have also responded below:

Reviewer #1

Minor corrections:

page 17 lines 401-404

“While these results need to be validated in a larger cohort and with various treatment schemes; this work builds upon the important work from Liu’s recent work, suggesting a more and less aggressive molecular subtype of RB – our work allows for detection of this from the AH in vivo.”

Grammar is not correct, and ref to “Liu’s recent work” to be included, as follows: “While these results need to be validated in a larger cohort and with various treatment schemes, this work builds upon the important work from Liu’s recent work REF, suggesting a more and less aggressive molecular subtype of RB. Our work allows for detection of subtype from the AH in vivo.”

Response: *Thank you for catching this! This sentence now in page 19 reads as the reviewer has correctly suggested: While these results need to be validated in a larger multi-center cohort with various treatment schemes, this work builds upon the important work from Liu et al.³⁶, suggesting a more and less aggressive molecular subtype of RB. Our work allows for detection of this subtype from the AH in vivo. Once validated prospectively, this has potential for direct impact to these young patients with RB by allowing the clinician to understand the state of the tumor and the likelihood of salvage with various therapies.*

Reviewer #2

(Remarks to the Author):

The authors have done an excellent job at addressing prior concerns. No additional concerns are noted. The study is novel and high impact, and in time could lead to improved care for RB patients based on AH cfCNA and epigenetic signatures. I recommend the paper be accepted.

Response: *Thank you for your time and expertise in reviewing our work.*

Reviewer #3

(Remarks to the Author):

While I agree that AH DNA methylation analysis may one day complement the other features of retinoblastoma that are used for staging, it is unlikely that it will ever stand alone because of the complexity of the disease. As noted, location of the primary tumor (near optic nerve, macula, vitreal seeds, anterior chamber), possibility of preserving vision

and family preference for preserving the eye in the absence of vision are all important considerations.

Response: *We agree. We have added 'combined with clinical features' to the following sentence on page 19. "Once validated prospectively, this has potential for direct impact to these young patients with RB by allowing the clinician to understand the state of the tumor, and **combined with clinical features**, the likelihood of salvage with various therapies."*

My major concern with this manuscript are:

1) feasibility of AH DNA methylation has been demonstrated and the next step in the field is a large prospective study through a cooperative group. This tiny study does not advance the field.

Response: *The authors agree with the reviewer completely that a large prospective, multi-site study is needed. This is in fact underway. We have **emphasized this on page 19**: "While these results need to be validated in a larger multi-center cohort with various treatment schemes, this work builds upon the important work from Liu et al.³⁶, suggesting a more and less aggressive molecular subtype of RB." **And on page 20**: "Although our analyses identified regions of interest that may help RB patients in the clinic, these findings are limited by small sample size from a single center. A prospective, multi-center collaborative effort with a large sample size of AH cfDNA from salvaged eyes of RB patients are needed to validate these findings regarding Cluster A and B and assess their clinical relevance with multiple therapeutic options."*

2) Claims that the underlying molecular mechanism can be inferred from DNA methylation are not supported by the data. While a small percentage of tumors may have hemizygous promoter hypermethylation, there is no treatment specific to those patients. Moreover, there are no druggable genetic or epigenetic targets in retinoblastoma despite extensive studies. Therefore, the two major claims of the paper are not supported by the data.

They have not demonstrated that DNA methylation can be used to stratify patients for treatment (study is much too small to change practice). They have not demonstrated any evidence that the DNA methylation data can be used to customize treatment.

Response: *The authors **do not claim** that currently this will change treatment for patients with RB, nor that we have identified a target. We claim that the AH liquid biopsy facilitates an in vivo analysis of methylation profiles of the tumor. This enables classification of these tumors into subtypes (either cluster A/B, or via Liu's work subtype 1/2) which are potentially prognostic, and in a **future state** may be clinically impactful. To ensure we are not overstating that there are any current treatment paradigms for RB that would change we have added '**future**' to therapy statements and clarified that these have not been used for RB". **On page 16**: "Taken together, these genes not only serve as prognostic biomarkers to predict eye salvage, tumor aggressiveness and likely response to treatment, but opens the door to **future applications of** predictive medicine by facilitating an in vivo evaluation of potential therapeutic targets for patients with RB, particularly those*

*with more aggressive disease (Figure 6 and Supplemental Table 2).” **On page 18:** “This opens a multi-omics approach to the AH enabling us to characterize the global methylation pattern of RB tumors in vivo at diagnosis and during therapy, thus obviating the need for tumor tissue. Moreover, unlike genetic alterations, RB1 epigenetic silencing is reversible and may be a therapeutic target of DNA methylation inhibitors. **These have not been used for the treatment of RB, but have been** used for treatment of several cancer types in clinical trials.^{28,53}”*

We thank the reviewers and the editors for their time and insightful responses in the review of this research.